# Spectral anomalies and broken symmetries in maximally chaotic quantum maps

**Laura Shou[1,2]⋆, Amit Vikram[3]† and Victor Galitski[3]‡**

**1** Condensed Matter Theory Center and Joint Quantum Institute, Department of Physics,
University of Maryland, College Park, MD 20742, USA
**2** School of Mathematics, University of Minnesota, Minneapolis, MN 55455, USA
**3** Joint Quantum Institute and Department of Physics,
University of Maryland, College Park, MD 20742, USA

⋆ lshou@umd.edu , † amitv@umd.edu , ‡ galitski@umd.edu

## Abstract

Spectral statistics such as the level spacing statistics and spectral form factor (SFF) are widely expected to accurately identify "ergodicity," including the presence of underlying macroscopic symmetries, in generic quantum systems ranging from quantized chaotic maps to interacting many-body systems. By studying various quantizations of maximally chaotic maps that break a discrete classical symmetry upon quantization, we demonstrate that this approach can be misleading and fail to detect macroscopic symmetries. Notably, the same classical map can exhibit signatures of different random matrix symmetry classes in short-range spectral statistics depending on the quantization. While the long-range spectral statistics encoded in the early time ramp of the SFF are more robust and correctly identify macroscopic symmetries in several common quantizations, we also demonstrate analytically and numerically that the presence of Berry-like phases in the quantization leads to spectral anomalies, which break this correspondence. Finally, we provide numerical evidence that long-range spectral rigidity remains directly correlated with ergodicity in the quantum dynamical sense of visiting a complete orthonormal basis.



# 1  Introduction

## 1.1  Background and motivation

The connection between the statistics of energy levels and a variety of ergodic phenomena is a foundational problem in the study of quantum signatures of chaos [1] and the statistical mechanics of quantum many-body systems [2]. In *generic* systems with a classical limit or many-body structure, an empirically successful approach has been to look for signatures of eigenvalue statistics associated with random matrix theory (RMT) [3], in order to diagnose "ergodicity" if these are present [4–20], and infer its absence otherwise [21–26]; indeed, the presence of "ideal" RMT statistics can be shown to be sufficient (but not necessary) for an ergodic exploration of an orthonormal basis in the Hilbert space of a general quantum system [27]. However, for a complete understanding of the utility of eigenvalue statistics, it is essential to quantitatively characterize deviations from this idealized behavior, particularly to identify where such statistics no longer accurately diagnose different forms of ergodicity/thermalization.

There are a number of interesting systems where deviations from RMT have been observed that point to an increasing need to characterize non-RMT behavior [21, 27–35]. Barring specific cases with alternate explanations, these deviations are generally due to *emergent* quantum symmetries not present in the classical system [1], usually connected to the classical periodic orbits, that lead to ergodicity-breaking after the Ehrenfest time [36, 37] at which classical and quantum evolutions diverge significantly. Prominent examples include the modular multipli-

cation [30, 38, 39] and cat maps [28, 29] (which become exactly periodic in their standard quantization), and chaotic dynamics on arithmetic domains in hyperbolic surfaces [32, 33], where RMT statistics are present only if specific boundary conditions are imposed on quantization [34], being strongly violated by emergent Hecke symmetries [32] otherwise.

In this work, we identify and characterize anomalies in spectral statistics of a different (and essentially opposite) nature to the above systems, originating in the quantum mechanical breaking of _discrete_ symmetries that are rigorously present at the macroscopic scale. The existence and relevance of such anomalies is suggested, for instance, by studies of certain exceptional billiard systems [40–43]. Specifically, we consider quantizations of maximally chaotic quantum maps in which we show that discrete macroscopic symmetries are _not_ accurately reflected in the most commonly used measures of spectral statistics: (1) the spectral form factor (SFF) [1, 9], which measures spectral rigidity over different energy scales as a function of time (namely, long-range at early times, and short-range at late times), (2) the (short-range) nearest neighbor level spacing statistics [1, 4–7], and (3) the adjacent gap ratios [44] (characterizing the short-range next-nearest-neighbor statistics). The short-range statistics in particular show especially stark violations. These violations are striking in the context of the use of spectral statistics to _identify_ discrete symmetries of the time evolution operator. While such diagnostics are effective in a variety of systems exhibiting block RMT behavior [10, 45–48], our results show they cannot always be relied upon, even in simple systems with a well-defined classical limit.

## 1.2 Summary of this paper

We aim to illustrate the unreliability of common spectral statistics in identifying discrete symmetries as may be present in quantized chaotic maps or many-body systems. To ensure that the systems being compared have an identical and well-understood macroscopic behavior, we consider classical maps that are _known_ to have two discrete symmetries that square to unity (i.e., restore the original system on acting twice). More specifically, we study spectral statistics in different quantizations [49, 50] of the $A$-baker's maps, which are classically paradigmatic examples of ergodicity with maximally chaotic (Bernoulli) behavior [51]. Incidentally, in addition to having a classical limit, these quantizations are particularly amenable to implementation as many-body Floquet quantum circuits [52, 53]. Further, all these quantizations reduce in the classical limit to the same classical $A$-baker's maps, and thus possess the same two discrete symmetries (Sec. 2). These are: a canonical reflection symmetry, and an anticanonical time-reversal symmetry, which respectively correspond to a unitary reflection and antiunitary time-reversal operator on quantization.

Our main qualitative results, described more thoroughly in Sec. 3, are as follows. While the spectral statistics of some of these quantizations are already known to be "unusual", a key observation in this work is that these unusual features can be satisfactorily organized in terms of a simple, and potentially generalizable, picture of different levels of "discrete symmetry breaking" in the spectral statistics. These anomalies are to be evaluated relative to the following general expectation based on RMT [1, 3]: the presence and absence of the time-reversal symmetry respectively correspond to COE and CUE level statistics, with the presence or absence of the reflection symmetry indicating a 2-block or 1-block structure of the associated random matrix. In particular, based on their classical symmetries, quantized $A$-baker's maps would be expected to have the spectral statistics of 2-block COE. With this context, we identify spectral anomalies of two types (Sec. 3.2, 3.3):

1. "Weak anomalies" whose primary effect is in the regime of long times corresponding to short-range energy spacings, leading to full (single-block) RMT-like behavior in the mean gap ratio statistic and nearest neighbor level spacings for large $A$ (ranging from 1-block

Figure 1: Visualization of the action of the 3-baker's map, starting from the left unit square and ending with the right unit square. The intermediate step shows the stretching, cutting, and stacking operation described by Eq. (1).

COE to CUE). However, the early-time SFF is consistent with the presence of the unitary reflection symmetry (2-block COE). These demonstrate that short-range measures can be misleading indicators of discrete symmetries.

2. "Strong anomalies" that affect even the regime of early times and long-range energy spacings in addition to the long-time regime as above, where even the early-time SFF shows RMT behavior consistent with the absence of unitary symmetries (1-block COE). These show that even long-range measures can be misleading in certain circumstances.

Subsequently, we study the connection between these spectral anomalies and dynamics. We show analytically and numerically that strong anomalies emerge from the inclusion of additional phases in specific quantizations [49], which have no impact in the classical limit, but occur as Berry-like phases in the semiclassical periodic orbit expression (Sec. 3.4). Further, we numerically study quantum dynamics in the Hilbert space in the sense of cyclic ergodicity [27], and find that strong anomalies appear to induce cyclic ergodicity where weak anomalies do not, verifying the direct connection between long-range spectral statistics and ergodic quantum dynamics irrespective of classical symmetries (Sec. 3.5). The remaining sections offer additional analytical and numerical details concerning these results.

## 2 Models

In this section, we introduce the classical and quantum systems.

**Classical maps**  The classical maps we consider are the $A$-baker's maps [49, 51, 54], which act on the 2-torus $\mathbb{T}^2 = \mathbb{R}^2/\mathbb{Z}^2$ (identified with the unit square) via

$$(q,p) \mapsto \left( Aq - \lfloor Aq \rfloor, \frac{p + \lfloor Aq \rfloor}{A} \right), \tag{1}$$

for $(q,p) \in [0,1) \times [0,1)$ and $A \geq 2$ an integer. When $A = 2$, this is the same as the usual baker's map. We depict the action of the $A$-baker's map for $A = 3$ on the unit square in Fig. 1. In what follows we may refer to $A$ as the "scaling factor" of the map.

The classical $A$-baker's map is equivalent to a 2-sided Bernoulli shift [49, 51, 55] and is thus maximally chaotic. It represents a fairly "universal" model of chaotic dynamics, as any K-mixing (ergodic *and* chaotic) system with sufficiently large Kolmogorov-Sinai entropy [56] (essentially the sum of nonnegative Lyapunov exponents) $h \geq \ln A$ can be coarse-grained into a given $A$-baker's map (or most directly, the corresponding Bernoulli shift), by the Sinai factor theorem [57,58]. The $A$-baker's maps possess two symmetries, a time-reversal (TR) symmetry $T : (q,p) \mapsto (p,q)$ and a reflection symmetry

$$R : (q,p) \mapsto (1-q, 1-p), \tag{2}$$

which will play key roles in our analysis. Due to the time-reversal symmetry, one expects the RMT symmetry class for the corresponding quantized systems to be that of the circular orthogonal ensemble (COE). Additionally, due to the reflection symmetry, one expects two distinct COE symmetry classes, leading to an overall behavior resembling a direct sum of two COE matrices. As we will demonstrate, however, these general expectations need not hold even approximately when the classical symmetries are broken upon quantization.

We briefly note that while we do not exhaustively verify the absence of any further classical symmetries for $A > 2$, numerical results counting values of the classical action on orbits (as defined in Eq. (30)) suggest the above symmetries are likely the only two. In any case however, the main conclusions of this paper that spectral statistics can fail to detect symmetries would still hold.

Here, we also emphasize the need to focus on "macroscopic" symmetries rather than the most general quantum symmetries for an individual system. In any quantum system with non-degenerate (quasi-)energy levels $|E_n\rangle$ that are eigenstates of a unitary map $\hat{U}$, unitary operators of the form $\hat{V}(v_n) = \sum_n v_n |E_n\rangle\langle E_n|$ exhaustively satisfy $[\hat{U}, \hat{V}(v_n)] = 0$, and represent the full set of unitary symmetry operations. This means that any nondegenerate quantum system has the same set of unitary quantum symmetries given by $\{\hat{V}(v_n)\}$, and it is not formally possible to directly associate spectral statistics with intrinsic quantum symmetries.[1] As described below, the conventional expectation in the literature [1, 2, 10] relies on a more ambiguous, but also more physically relevant, approach.

This more practical approach is to consider only the subset of symmetry operations $\hat{V}(\bar{v}_n)$ that satisfy some "physicality" criteria corresponding to a macroscopic limit, such as having a well-defined classical limit [1, 10], or being sufficiently "accessible" (e.g., generated by few-body observables or their simple algebraic combinations) in a many-body system [2]. This may also include symmetry operators that appear diagonal or "sufficiently" simple in a basis that remains relevant in a macroscopic limit. It is with respect to such "macroscopic" symmetries that the empirical connection between symmetries and spectral statistics can be formulated for an individual quantum system. Consequently, in the absence of a nontrivial and unambiguous way to define intrinsic quantum symmetries, we must focus on the veracity of the correspondence between the symmetry operators corresponding to *macroscopic* symmetries (identified through a classical limit in our context of $A$-baker's maps) and different measures of spectral statistics. We also note that several operators can have the same classical limit (in the semiclassical context, due to terms that vanish as $\hbar \to 0$), and we may have to restrict to "obvious" or sufficiently simple symmetry operators. Even with this restriction, we show that the correspondence can be highly nontrivial for relatively simple quantum maps.

**Balazs–Voros, Saraceno, and generic quasiperiodic quantizations**   For quantizing a map like the classical $A$-baker's map Eq. (1), there is no unique method; essentially one just requires the associated quantum map to be a unitary $N \times N$ matrix that reduces to the classical map in the semiclassical limit $N \to \infty$. The first quantization of the baker's map was given by Balazs and Voros in [49]; for the simplest case $A = 2$ and $N$ even, this reads

$$\hat{B}_N = \hat{F}_N^{-1} \begin{pmatrix} \hat{F}_{N/2} & \mathbf{0} \\ \mathbf{0} & \hat{F}_{N/2} \end{pmatrix}, \tag{3}$$

---

[1]This issue is often circumvented in approaches based on random matrix *ensembles* [1, 45, 46, 48] with a block structure representing symmetry sectors. This is because each member of such an ensemble has a different eigen-basis $|E_n\rangle$, and the only symmetries that apply to every element in the ensemble are those that respect the block strucutre, providing a way to justify ignoring most of the intrinsic quantum symmetries. However, this reasoning cannot be transplanted to individual systems with a uniquely specified eigenbasis $|E_n\rangle$.

where $\hat{F}_N$ is the $N \times N$ discrete Fourier transform (DFT) matrix defined via

$$(\hat{F}_N)_{jk} = \frac{1}{\sqrt{N}} e^{-2\pi i jk/N}, \qquad j,k = 0, \ldots, N-1 .$$

This quantization using the standard DFT matrix is associated with periodic boundary conditions on the torus $\mathbb{T}^2$. In order to study different quantum symmetries, we will consider the natural "generic" quantization for the $A$-baker's map with quasiperiodic boundary conditions [49,50,59] corresponding to $\theta = (\theta_1, \theta_2) \in [0,1)^2$,

$$\text{Gen}_{A,N}^{\theta_1,\theta_2} = (\hat{F}_N^{\theta_1,\theta_2})^{-1} \bigoplus_{j=0}^{A-1} \hat{F}_{N/A}^{\theta_1,\theta_2} , \tag{4}$$

where

$$(\hat{F}_N^{\theta_1,\theta_2})_{jk} = \frac{1}{\sqrt{N}} e^{-2\pi i(j+\theta_1)(k+\theta_2)/N} , \tag{5}$$

is a generalized DFT matrix, and $N \in A\mathbb{N}$. The direct sum part of Eq. (4) produces a block diagonal matrix consisting of generalized DFT matrices $\hat{F}_{N/A}^{\theta_1,\theta_2}$.

The case $\theta_1 = \theta_2 = 0$ is the Balazs–Voros quantization of the $A$-baker's map, for which we may use the abbreviated label "BV" in plots or tables. The Balazs–Voros quantizations preserve an operator TR symmetry but break an operator reflection symmetry (Sec. 4.1), and for $A = 2$ were observed to show anomalous level spacings behavior depending on the dimension $N$ [49].

The case $\theta_1 = \theta_2 = 1/2$ is the Saraceno quantization from [50], and corresponds to antiperiodic boundary conditions. This quantization preserves TR symmetry and moreover preserves the classical reflection symmetry, as it commutes with the microscopic reflection operator $R_N : |x\rangle \mapsto |N-x-1\rangle$. As a consequence, this was observed for $A = 2$ to restore COE level spacing statistics within each symmetry class.

In general, for $\theta_1 \neq \theta_2$, the generic quantization in Eq. (4) does not appear to preserve a clear operator TR or reflection symmetry like in the Saraceno case. Possible symmetries are discussed further in Sec. 4.1 and Appendix A.

**Shor baker quantization**  In addition to the above generic (quasi)periodic quantizations, we consider the "Shor baker quantizations" from [38,39]. These quantizations are part of the quantum baker's map decomposition of the modular multiplication operator in Shor's factoring algorithm [30], and can be defined as

$$\hat{S}_{A,N} = \hat{F}_N^{-1} \left( \bigoplus_{j=0}^{A-1} e^{2\pi i j^2/A} \hat{F}_{N/A}^{0,-j/A} \right), \tag{6}$$

where $F_{N/A}^{0,-j/A}$ denotes a generalized DFT matrix defined via Eq. (5). These Shor baker quantizations again appear to break both the operator TR and reflection symmetries.

**Phase variants**  Finally, we consider "phase variant quantizations" by adding arbitrary Berry-like phases $e^{2\pi i\alpha} = (e^{2\pi i\alpha_0}, \ldots, e^{2\pi i\alpha_{A-1}})$ to the DFT block sectors of the previous $A$-baker's map quantizations. These are written in the right column of Tab. 1. These quantizations have historically been considered as variations on the usual Balazs–Voros or Saraceno quantizations since [49], but generally are overlooked in favor of the simpler standard/phaseless quantizations. For generic or random phases, we will see that the phase variant quantizations exhibit significantly different spectral statistics than their corresponding standard/phaseless quantizations.

Table 1: Definitions of the different quantizations of the $A$-baker's map. Balazs–Voros is the same as $\text{Gen}_A^{0,0}$, and Saraceno the same as $\text{Gen}_A^{1/2,1/2}$. The "default" quantizations will be the standard/phaseless ones, and we may simply refer to them as the "Balazs–Voros/Saracneo/Generic/Shor baker" quantizations, while for the quantizations with arbitrary phases $e^{2\pi i\alpha}$ we will always specify that it involves the extra phases.

| | Standard/Phaseless | Phase variant |
|---|---|---|
| Balazs–Voros | $\hat{F}_N^{-1} \bigoplus_{j=0}^{A-1} \hat{F}_{N/A}$ | $\hat{F}_N^{-1} \bigoplus_{j=0}^{A-1} e^{2\pi i\alpha_j} \hat{F}_{N/A}$ |
| Saraceno | $\left(\hat{F}_N^{\frac{1}{2},\frac{1}{2}}\right)^{-1} \bigoplus_{j=0}^{A-1} \hat{F}_{N/A}^{\frac{1}{2},\frac{1}{2}}$ | $\left(\hat{F}_N^{\frac{1}{2},\frac{1}{2}}\right)^{-1} \bigoplus_{j=0}^{A-1} e^{2\pi i\alpha_j} \hat{F}_{N/A}^{\frac{1}{2},\frac{1}{2}}$ |
| Generic $\text{Gen}_A^{\theta_1,\theta_2}$ | $(\hat{F}_N^{\theta_1,\theta_2})^{-1} \bigoplus_{j=0}^{A-1} \hat{F}_{N/A}^{\theta_1,\theta_2}$ | $(\hat{F}_N^{\theta_1,\theta_2})^{-1} \bigoplus_{j=0}^{A-1} e^{2\pi i\alpha_j} \hat{F}_{N/A}^{\theta_1,\theta_2}$ |
| Shor baker | $\hat{F}_N^{-1} \bigoplus_{j=0}^{A-1} e^{2\pi ij^2/A} \hat{F}_{N/A}^{0,-\frac{j}{A}}$ | $\hat{F}_N^{-1} \bigoplus_{j=0}^{A-1} e^{2\pi i\alpha_j} \hat{F}_{N/A}^{0,-\frac{j}{A}}$ |

All of the quantizations in Tab. 1 are quantizations of the classical $A$-baker's map in the sense that they map coherent states localized in phase space near $(q, p)$, to coherent states localized in phase space near the classical time-evolved point $\left(Aq - \lfloor Aq \rfloor, \frac{p+\lfloor Aq \rfloor}{A}\right)$ as $N \to \infty$. For details, see [60, §4] and [39, Suppl. Mat.], noting that for quasiperiodic boundary conditions one must use the appropriate quasiperiodic coherent states and generalized DFT matrix $\hat{F}_N^{\theta_1,\theta_2}$. Additionally, for the Balazs–Voros (and Saraceno) quantizations, the argument in [60, §5] proves a rigorous "Egorov property" concerning time-evolution of quantum observables $\text{Op}_N(a)$ corresponding to classical observables $a$ on $\mathbb{T}^2$ supported away from classical discontinuities,

$$\|\hat{U}_N^t \, \text{Op}_N(a)\hat{U}_N^{-t} - \text{Op}_N(a \circ B^{-t})\| \xrightarrow{N\to\infty} 0\,, \tag{7}$$

where $\hat{U}_N$ is the quantization and $B$ is the classical $A$-baker's map. The argument is insensitive to phases on the DFT blocks, so that the same rigorous correspondence holds for their corresponding phase variant quantizations. We expect the same argument (with some minor adaptations) applies to the generic quasiperiodic and Shor baker quantizations both with and without phases.

## 3 Results

### 3.1 Overview of results

In this section, we explain the main results summarized in Tab. 2, which compares the nearest-neighbor level spacing statistics and spectral form factor behavior by quantization and presence of quantum symmetries. We provide the numerical results for the level spacing statistics, and both analytical and numerical results for the early time SFF slope. Due to the classical TR and reflection symmetries of the classical $A$-baker's map, one would expect its quantizations to exhibit spectral statistics similar to a 2-block COE matrix (a direct sum of two independent, equal sized COE matrices). As has been well-known since [49], this already does not hold for the level spacing statistics of the Balazs–Voros quantization with $A = 2$, which display intermediate level spacing statistics due to the mixing of symmetry sectors. But as we will see, there are several subtleties involved with the spectral statistics, and the results will depend

on both the spectral statistic chosen and the particular quantization type. We emphasize the following main points.

(A) Unlike the $A = 2$ case, for large $A$, the level spacing statistics actually do appear to follow classical RMT behavior for all considered quantizations. However, this RMT behavior can be of the wrong symmetry class (e.g. CUE vs COE) and/or reflect the wrong number of symmetry sectors.

(B) For the standard/phaseless quantizations, the early time SFF behavior correctly identifies the RMT symmetry class and symmetry sectors, even as the level spacings do not. This provides a resolution for the non-RMT level spacing statistics in [49], as well as for the wrong symmetry class behavior in the aforementioned point. Such spectral anomalies, where the long-range statistics remain reliable even as the short-range ones do not, are those we term "weak anomalies", and they appear to be well-described by a *block* Rosenzweig–Porter-like interpolation between RMT ensembles.

(C) The Berry-like phases in the phase variant quantizations produce "strong spectral anomalies", where even the early time SFF misses one of the classical symmetries. Using a semiclassical periodic orbit analysis, we analytically characterize the early time SFF slope as a function of the phase choices, and show a generic choice of phases (probability 1 set) will always lead to strong anomalies. We note that the reflection and TR symmetries continue to emerge in the classical limit despite these phases.

(D) The presence of strong anomalies is verified numerically to be tied to ergodicity in a quantum dynamical sense of exploring an orthonormal basis in the Hilbert space [27], irrespective of symmetries in the classical limit. However, weak anomalies do not appear to be sufficiently strong to induce ergodic dynamics in this sense.

## 3.2 Nearest-neighbor level spacing statistics

The nearest-neighbor level spacings statistics of an $N \times N$ unitary matrix are obtained by ordering the eigenangles $\theta_i$, and considering the normalized nearest-neighbor spacings (or gaps)

$$s_i = \frac{N}{2\pi}(\theta_{i+1} - \theta_i), \qquad i \in \mathbb{Z}/N\mathbb{Z}. \tag{8}$$

The distribution of the $(s_i)$ can then be directly compared to those of classical RMT ensembles. A useful single statistic computed from the level spacings is the mean (adjacent) gap ratio statistic from [44], given by $\langle \tilde{r} \rangle = \left\langle \min\left(\frac{s_{i+1}}{s_i}, \frac{s_i}{s_{i+1}}\right) \right\rangle_i$, where the average is over all $i \in \mathbb{Z}/N\mathbb{Z}$. This statistic provides a single value that can be used to compare the closeness to RMT level spacings, and does not require any normalization or unfolding of the eigenvalues [44]. For reference, the mean gap ratio values for the RMT ensembles as derived in [48,61] are provided in Tab. 3.

While RMT level spacing statistics are commonly used as an indicator (or even definition) of "quantum chaotic" systems [1], the $A$-baker's map quantizations can exhibit non-universal level spacing statistics that are strongly sensitive to the particular quantization choice. The first hint of complication is that the Balazs–Voros quantization in Eq. (3) ($A = 2$) was observed in [49] to have level spacing statistics that vary depending on $N$; they almost never look COE or block COE, which was explained as due to the quantization breaking the classical reflection symmetry in Eq. (2) and mixing symmetry sectors together.

Table 2: Summary of spectral statistics for the various quantizations of the $A$-baker's map, with the standard or phaseless quantizations in the top section, and the random phase variant quantizations in the bottom section. The columns for the generic quantization $\text{Gen}_A^{\theta_1,\theta_2}$ and its phase quantization reflect the choices $\theta = (0.2, 0.7)$ and $(0, 0.5)$ for numerics, though the SFF slope behavior we derive through the periodic orbit analysis applies to any choice of $\theta$. As seen in the table, the level spacing statistics vary greatly across all quantizations, and only accurately reflect the classical symmetry sectors over all $A$ for the standard Saraceno quantization, which preserves both classical symmetries upon quantization. Ruling out preserved classical symmetries has subtleties however (cf. Sec. 4.1), and the rows describing preserved classical symmetries correspond specifically to ruling out standard or "obvious" quantum symmetries, as well as to identifying whether the symmetry is reflected in the short range spectral statistics. For long range statistics, the early time SFF slope successfully identifies the symmetry sectors for all standard/phaseless quantizations, even when the operator does not exhibit a clear analogue of the classical symmetries. However, the SFF misses the reflection symmetry in the random phase variant quantizations in the bottom section. The entries labeled "mixed" indicate level spacings that do not adhere to a single RMT or block-RMT ensemble, and instead look somewhere inbetween ensembles.

| | | BV | Saraceno | $\text{Gen}_A^{\theta_1,\theta_2}$ | Shor baker |
|---|---|---|---|---|---|
| Preserved | TR | Y | Y | N | N |
| (classical) sym. | Reflection | N | Y | N | N |
| $A = 2$ | Level spacings | mixed | 2-COE | 2-COE/mixed | mixed |
| | SFF slope | 4 | 4 | 4 | 4 |
| $A$ large | Level spacings | COE | 2-COE | CUE | CUE |
| | SFF slope | 4 | 4 | 4 | 4 |

| | | BV($\alpha$) | Saraceno($\alpha$) | $\text{Gen}_A^{\theta_1,\theta_2}(\alpha)$ | Shor baker($\alpha$) |
|---|---|---|---|---|---|
| Preserved | TR | Y | Y | N | N |
| (classical) sym. | Reflection | N | N | N | N |
| $A = 2$ | Level spacings | COE | COE | COE/mixed | mixed |
| | SFF slope | 2 | 2 | 2 | 2 |
| $A$ large | Level spacings | COE | COE | CUE | CUE |
| | SFF slope | 2 | 2 | 2 | 2 |

Surprisingly, as demonstrated by Figs. 2 and 3, we find the level spacing statistics and mean gap ratio statistic for the higher scaling factor $A$-baker's maps begin to look very close to those of a single COE or CUE matrix as $A$ increases, for all quantizations except the standard Saraceno quantization. Thus for large values of $A$, these level spacing statistics appear RMT, but reflect the *wrong* symmetry classes. The effect of the classical reflection symmetry appears to completely disappear for large $A$ (for non-Saraceno quantizations), and for some quantizations the TR symmetry separating COE from CUE is ignored as well.

Although all quantizations share the classical limit of an $A$-baker's map, they exhibit a wide variety of level spacing and gap ratio statistics, ranging from the expected 2-block COE behavior, to single block COE, to single block CUE, and to intermediate or mixed statistics inbetween two RMT ensembles. We observe that for large $A$, it appears these short-range spectral statistics reflect certain symmetries of the quantized operator (Sec. 4.1, Appendix A), but not necessarily those of the underyling classical map.

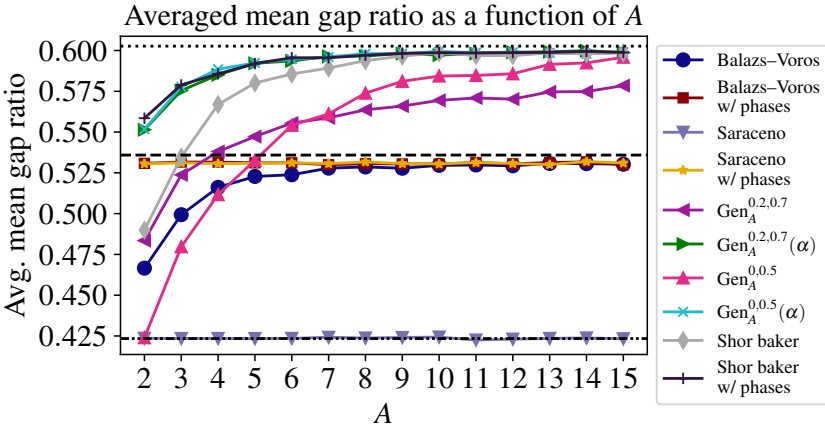

Figure 2: Averaged mean gap ratio for quantizations of the $A$-baker's map, as a function of $A$. Each point represents an average over 50 values of $N \in A\mathbb{N}$, starting near $N = 5000$. The horizontal lines, from top to bottom, plot the RMT reference values for CUE (dotted), COE (dashed), and 2-block COE (dash-dot-dotted). Only the standard Saraceno quantizations (downward triangle) exhibit mean gap ratios close to the expected 2-block COE value for all $A$.

## 3.3 Spectral form factor

The spectral form factor (SFF) is the Fourier transform of the 2-point level correlation function [1,3]. For $\hat{U}_N$ an $N \times N$ unitary matrix, the SFF is given by the formula

$$\text{SFF}(t) = \frac{1}{N}|\text{Tr}(\hat{U}_N^t)|^2 = \frac{1}{N}\sum_{j,k=1}^{N} e^{it(\theta_j - \theta_k)}, \tag{9}$$

where $(\theta_j)_{j=1}^{N}$ are the eigenangles of $\hat{U}_N$. The normalization is chosen so that the SFF can be conveniently analyzed and compared across different values of $N$. For early times $t > 0$, the SFF measures long-range spectral correlations, while for larger times $t$, the SFF describes finer spectral correlations such as level spacings and eventually discreteness of the spectrum.

Letting $\tau = t/N$, there is the well-known formula for the COE form factor averaged over the random ensemble in the limit $N \to \infty$ [1], which for early times $\tau$ yields $\langle\text{SFF}_{\text{COE}}(\tau)\rangle = 2\tau + \mathcal{O}(\tau^2)$. For 2-block COE matrices, the corresponding ensemble-averaged SFF is $\langle\text{SFF}_{\text{2-COE}}(\tau)\rangle = \langle\text{SFF}_{\text{COE}}(2\tau)\rangle$. Thus the early time SFF slope is 2 for a single COE matrix, and 4 for the 2-block COE matrix. For the $A$-baker's map quantizations, since we do not have an ensemble of matrices to average over, we average the SFF by averaging over neighboring points as described in Appendix C.

We first demonstrate that the early time (averaged) SFF resolves the two issues with the level spacing statistics for the standard/phaseless quantizations, (i) the non-universal behavior for small $A$ of the Balazs–Voros/Generic/Shor baker quantizations, and (ii) the apparent disappearance of two distinct symmetry sectors for the same quantizations with larger $A$. These cases thus correspond to "weak anomalies", for which the SFF provides a satisfactory diagnostic of the spectral behavior and classical symmetries.

As shown in the top row of Fig. 4, for very early times $\tau$, the SFFs for the standard phaseless quantizations follow the slope 4 reference SFF behavior for the 2-block COE, correctly reflecting the classical map symmetries. The longer time behaviors (corresponding to shorter range statistics) however vary greatly. For larger $\tau$, the Saraceno quantizations (and $\text{Gen}_{A=2}^{0,0.5}$) continue to follow the 2-block COE SFF, as previously demonstrated for the Saraceno $A = 2$

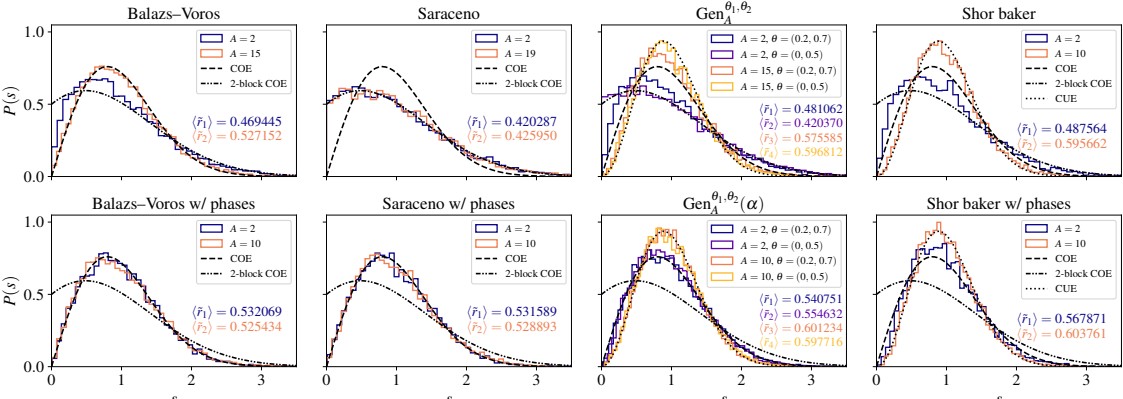

Figure 3: Level spacing histograms for the different quantizations of the $A$-baker's map, for $N = 9690$. Note the variety of behaviors–COE, 2-block COE, mixed/indeterminate, and even CUE–that can arise, despite the same classical map symmetries. Only the phaseless Saraceno quantizations and phaseless $\text{Gen}_{A=2}^{0,0.5}$ quantization appear to follow the 2-block COE curve. The mean gap ratio statistic $\langle \tilde{r} \rangle$ is also computed for each quantization.

quantization in [62], but the other standard quantizations appear to cross over to the single COE or CUE SFF at a time $\tau$ that decreases as $A$ increases. Since the level spacing statistics are short-range, corresponding to larger $\tau$, this faster cross-over explains the Balazs–Voros/Generic/Shor baker matrix level spacing histograms approaching those of a single COE or CUE matrix as $A$ increases. For these cases, which describe "weak anomalies", both the RMT nature and symmetry sectors are readily apparent in the SFF, in contrast to the differing information from the level spacing statistics.

The phase variant quantizations hold a surprise however. As shown in the bottom row of Fig. 4, the addition of random phases to the quantizations interferes with the classical reflection symmetry in a way that the early time SFF fails to detect it. Instead, the early time aeveraged SFF has slope 2, capturing only the classical TR symmetry. (The level spacings are of even less help, as seen in Figs. 2 and 3). We remark that from Fig. 4, it is not entirely clear whether it is the TR or reflection symmetry that is missed by the early time SFF; the SFF for several of the quantizations follows the COE curve which strongly suggests it is the reflection symmetry that is broken in those cases, but the SFF for other quantizations crosses over to the CUE curve. From the periodic orbit analysis below, we will see that it is still the reflection symmetry that is broken at early times in all cases. From the periodic orbit analysis, we will also be able to identify the specific phases $\alpha$ that produce an SFF slope of 4, which is a measure zero set but contains more elements than just those corresponding to the standard/phaseless ($\alpha_j = 0$) quantizations.

In addition to the SFF plots in Fig. 4, we plot the best fit SFF slope over a wide range of dimensions $N$ in Fig. 5. Unlike the standard/phaseless quantizations which produce SFFs with slope near 4 that accurately describe the classical symmetry sectors, the quantizations with random phases produce SFFs with slope near 2, thereby hiding the classical $R$ symmetry.

Overall, as summarized in Tab. 2, although the early time SFF slope correctly identifies both classical symmetries for the standard/phaseless quantizations ("weak anomalies"), it only captures one classical symmetry for the phase variant quantizations ("strong anomalies"). Meanwhile the level spacings fare worse, missing either one or both classical symmetries in almost all quantizations.

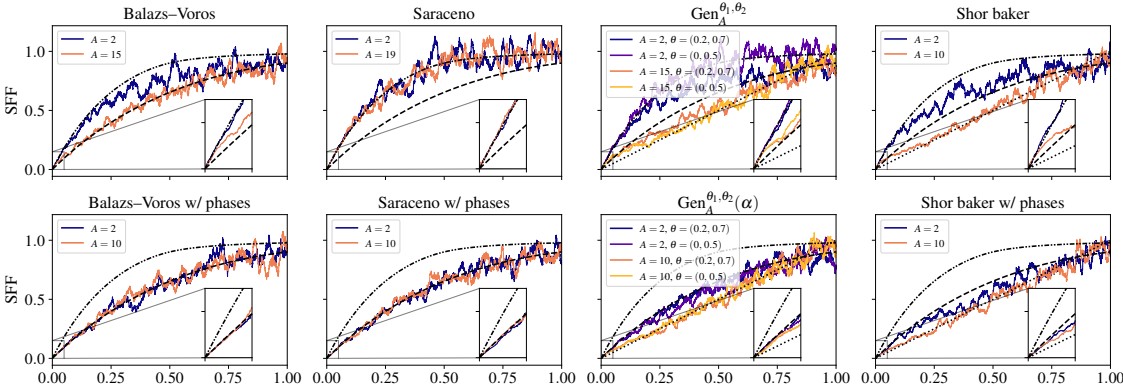

Figure 4: Averaged SFFs for the different quantizations of the $A$-baker's maps, for $N = 9690$. In the top row of standard phaseless quantizations, the very early time SFF follows the 2-block COE behavior (slope 4 at the origin), while for the bottom row of phase variant quantizations, the early time SFF has slope 2. All insets show up to $\tau = 0.05$, or up to $t = 484$ for $N = 9690$. For several of the larger $A$ quantizations, the transition away from the early time ($\tau \approx 0$) SFF slope behavior is already visible in this window. In general, the larger time SFF, corresponding to shorter range spectral statistics like the level spacings, vary greatly depending on the particular quantization.

Based on the level spacings and SFF behaviors, we find it appears that the spectral statistics for these quantized $A$-baker's maps look like those of a Rosenzweig–Porter-like [45] interpolation between a 2-block COE matrix and a standard CUE or COE matrix (for standard quantizations), or between a COE matrix and a CUE matrix (for phase variant quantizations). For the former case, this type of *block* Rosenzweig–Porter model was introduced (for block GUE) in [26] as a model for glassy behavior. In our case with unitary matrices, we will utilize a different interpolation to preserve unitarity, namely a geodesic path between unitary matrices $U_0$ and $U_1$ given by

$$f(t) = U_0 \exp(t \log(U_0^\dagger U_1)), \tag{10}$$

for $0 \leq t \leq 1$. For interpolating between a 2-block COE matrix $U_0$ and a COE matrix $U_1$, we write $U_0 = V^T V$ and $U_1 = W^T W$ for unitaries $V$ and $W$, apply Eq. (10) to obtain an interpolation $f_{VW}(t)$ between $V$ and $W$, and then take the interpolation $F(t) = f_{VW}(t)^T f_{VW}(t)$ between $U_0$ and $U_1$. In the other two cases, interpolating between 2-block COE and CUE or between COE and CUE, we just take $F(t)$ to be the same as $f(t)$ in Eq. (10). We plot the resulting level spacing statistics and SFF of the intermediate matrices $F(t)$ for different values of $t$ in Fig. 6, which show similar behavior as the statistics shown in Fig. 3 and 4.

## 3.4 Periodic orbit expansion

We now briefly analytically explain the above numerical observations for the early time SFF slope using a semiclassical periodic orbit expansion for the SFF of the $A$-baker's map quantizations [59, 62, 63], leaving the full details for Sec. 6. This analysis fills in the SFF slope values for the entirety of Tab. 2, and moreover identifies the precise measure zero set of phases $\alpha$ that lead to an SFF slope of 4 rather than 2. The slope 2 results we obtain for the specific models here differ from the usual periodic orbit theory expectation for generic systems, where one expects the early time SFF slope to faithfully reflect the number of symmetry sectors of the classical system [10, 47]. For the Saraceno quantization, which ends up as part of the measure zero set leading to the slope of 4, the SFF slope of 4 was derived in [62]. In the models

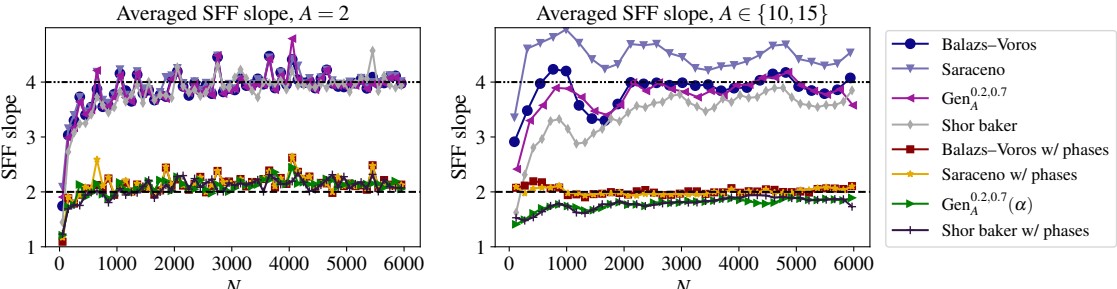

Figure 5: Averaged best fit early time SFF slope for $A = 2$ (left) and $A \in \{10, 15\}$ (right, $A = 15$ for standard Balazs–Voros, Saraceno, and $\text{Gen}_{A=15}^{0.2,0.7}$ quantizations, and $A = 10$ for the remaining). The quantizations with random phases show a slope near 2, while those without show a slope near 4. Some of the quantizations shown share the random choice of phases. Outliers where the least squares fitting had large error were removed prior to averaging (cf. Fig. 10, Appendix C).

here, the addition of phases alters the semiclassical trace formula as seen below, which can produce the SFF slope of 2. For the Shor baker quantizations, complications also arise due to the different generalized DFT blocks. This requires a more complicated analysis of the $t$-step propagator (Sec. 6.3, Appendix D), which we determine using coherent state evolution, in order to derive the corresponding trace formula.

In all of the following, $t \in \mathbb{N}$, and $N$ will be a multiple of $A^t$ for convenience. As we are interested in the SFF slope for early times $\tau = t/N$ as $N \to \infty$, we will assume $t \to \infty$ slowly, such as at a rate $\sim \log_A N$ or slower (so that $N$ can still be a multiple of $A^t$); this corresponds to $\tau \to 0$. We start with a matrix $\hat{U}_N = \hat{U}_N(\alpha)$ from the Generic phase variant quantization $\text{Gen}_A^{\theta_1,\theta_2}(\alpha)$ (which includes the Balazs–Voros and Saraceno phase quantizations),

$$\hat{U}_N = (\hat{F}_N^{\theta_1,\theta_2})^{-1} \bigoplus_{j=0}^{A-1} e^{2\pi i \alpha_j} \hat{F}_{N/A}^{\theta_1,\theta_2} \,. \tag{11}$$

One can readily check that applying the semiclassical propagator and saddle point method described in [63] (see also [62] and Eq. (D.2)) with these phases yields the periodic orbit approximation for $N \to \infty$,

$$\text{tr}\,\hat{U}_N^t \approx \sum_{\nu=0}^{A^t-1} \frac{1}{A^{t/2}} e^{2\pi i N S_\nu} e^{2\pi i \sum_{j=0}^{A-1} \alpha_j \eta_j(\nu)} \,, \tag{12}$$

where $S_\nu := \frac{\nu \bar{\nu}}{A^t-1}$ is the classical action, $\bar{\nu}$ is the (length $t$) base $A$ reversal of $\nu$, and $\eta_j(\nu)$ is the number of $j$'s in the (length $t$) base $A$ expansion of $\nu$. To estimate the SFF $\frac{1}{N}|\text{tr}\,\hat{U}^t|^2$, one expands Eq. (12) in a double sum over indices $\nu, \sigma$, and takes the "diagonal approximation" [9] with symmetry factors: The two classical symmetries are time-reversal $\nu \mapsto \bar{\nu}$ and reflection $R(\nu) = A^t - 1 - \nu$. Only summing over the orbits $\sigma \in \{\nu, \bar{\nu}, R(\nu), R(\bar{\nu})\}$ and their cyclic rotations results in

$$\frac{1}{N}|\text{tr}\,\hat{U}_N^t|^2 \approx \frac{2t}{N} + \frac{2t}{NA^t} \left( \sum_{j=0}^{A-1} e^{2\pi i(\alpha_j - \alpha_{A-1-j})} \right)^t \,. \tag{13}$$

We find that in order for the second term of Eq. (13) not to decay as we average over $t \to \infty$, we must have $\alpha_j = \alpha_{A-1-j}$ (modulo 1) for all $j$. Thus we obtain an SFF slope of 4 in this case, and a slope of 2 in all other cases. The requirement $\alpha_j = \alpha_{A-1-j}$ preserves a kind of "block"

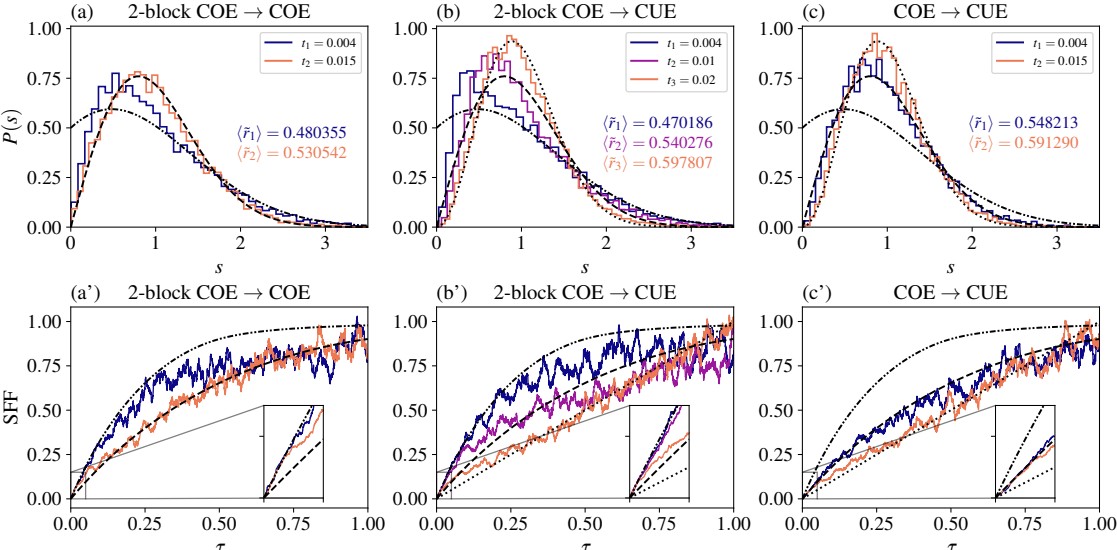

Figure 6: Level spacing histograms (top row) and SFFs (bottom row) for random instances of the Rosenzweig–Porter-like interpolation $F(t)$ (defined in the paragraph below Eq. (10)), for $N = 9690$. Each column involves two or three independent random matrices $F(t)$, one chosen for each $t$ value. Different values of $t$ appear to describe the various level spacings and SFF behaviors seen in Figs. 3 and 4.

$R$-symmetry, even though in general such quantizations can break the microscopic $R$-symmetry $|x\rangle \mapsto |N - 1 - x\rangle$.

The standard phaseless quantizations here have $\alpha_j = 0$ for all $j$, and thus meet the requirement for an SFF slope of 4, in agreement with numerics. We also note that when $\alpha_j = \alpha_{A-1-j} + 1/2 \pmod 1$, the approximation in Eq. (13) gives the value *zero* for the SFF at odd times $t$. In fact, this is exact for the Saraceno phase variant with these phases: When $\alpha_j = \alpha_{A-1-j} + 1/2$, then the resulting Saraceno $\hat{U}_N(\alpha)$ *anticommutes* with the reflection operator $R_N : |x\rangle \mapsto |N - 1 - x\rangle$, so that every eigenvalue $e^{i\lambda}$ comes with a partner $-e^{i\lambda}$, and $\operatorname{tr}\hat{U}_N^t = 0$ for odd $t \in \mathbb{N}$.

We now consider the Shor baker phase variant quantizations. Unlike the $\operatorname{Gen}_A^{\theta_1,\theta_2}(\alpha)$ phase variant quantizations, we recall this quantization involves different generalized DFT matrices for each block,

$$\hat{U}_N = \hat{F}_N^{-1}\left(\bigoplus_{j=0}^{A-1} e^{2\pi i \alpha_j} \hat{F}_{N/A}^{0,-j/A}\right). \tag{14}$$

In order to estimate the SFF using the periodic orbit expansion, we must first identify the correct $t$-step quantization $\hat{U}_N^{(t)}$ corresponding to this $\hat{U}_N$, which is complicated by the different generalized DFT blocks. By determining the behavior of $\hat{U}_N(\alpha)$ on maximally localized coherent states (Sec. 6.3), we can find the corresponding $t$-step propagator in mixed momentum-position basis (Eq. (39)), which is used to derive the trace formula (Eq. (41)),

$$\operatorname{tr}\hat{U}^{(t)} \approx \sum_{\nu=0}^{A^t-1} \frac{1}{A^{t/2}} e^{2\pi i N S_\nu} e^{\frac{2\pi i \nu \bar{\nu}}{A^t(A^t-1)}} e^{-2\pi i \frac{\phi(\nu)}{A}} e^{2\pi i \sum_{j=0}^{A-1} \alpha_j \eta_j(\nu)},$$

where $\phi(\nu) = -\sum_{j=2}^{t} a_j \sum_{i=1}^{j-1} a_i A^{-j+i}$. As calculated in Sec. 6.3, the extra factors in the trace

formula, with the diagonal approximation, eventually yield

$$\frac{1}{N}|\operatorname{tr}\hat{U}_N^t|^2 \approx \frac{2t}{N} + \frac{2t}{NA^t}\left(\sum_{j=0}^{A-1} e^{2\pi i(\alpha_j - \alpha_{A-1-j} + 2j/A)}\right)^t e^{2\pi it/A}. \tag{15}$$

Similar analysis then shows we obtain an averaged SFF slope of 4 iff

$$\alpha_{A-1-j} = \alpha_j + \frac{2j+1}{A} \mod 1, \qquad j \in [\![0:A-1]\!], \tag{16}$$

and slope 2 in all other cases. For the standard Shor baker quantization, $\alpha_j = j^2/A$, which satisfies Eq. (16). Unlike the condition on phases for the Balazs–Voros, Saraceno, and generic quasiperiodic quantizations, this condition does not seem to exhibit a clear "block" $R$-symmetry to mirror the classical one.

### 3.5 Symmetry breaking and quantum dynamical ergodicity

Having demonstrated that measures of spectral statistics can be incompatible with classical symmetries, we now consider the direct relation between spectral statistics and quantum dynamics in the Hilbert space. This is especially of interest in illustrating the fully quantum mechanical role of spectral anomalies or deviations from ideal random matrix behavior, irrespective of symmetries in the classical limit. We will take advantage of the distinct behavior of each measure across different quantizations of the $A$-baker's maps to contrast the role of short-range and long-range spectral statistics in influencing quantum dynamics. In particular, we will provide numerical evidence that long-range symmetry breaking or strong anomalies are sufficient to induce ergodicity (in a sense to be clarified below) in the quantum dynamics of the system, while short-range or weak anomalies have a milder effect that may not be significant in the $N \to \infty$ limit.

For this purpose, we will consider the notion of quantum cyclic ergodicity in the Hilbert space, introduced in Ref. [27] as a direct quantum dynamical counterpart to spectral statistics. There, it was shown that the presence of sufficient long-range spectral rigidity is tied to the existence of an orthonormal basis $\{|C_k\rangle\}_{k=0}^{N-1}$ where every initial state "visits" every other state in a cyclic sequence. This form of ergodicity is appropriate for time-independent unitary systems with (quasi-)energy conservation, and differs from more direct forms related to classical ergodicity possible in open or time-dependent quantum systems [64–66]. Quantitatively, the overlap of an initial state $|C_k\rangle$ with $|C_{k+t}\rangle$ after $t$ time-steps, called the persistence,

$$z_k^2(t) \equiv |\langle C_{k+t}|\hat{U}_N^t|C_k\rangle|^2, \tag{17}$$

must be larger than a cutoff $\eta^2(N) = cN^{-1}$ (where $c$ is some $\Omega(1)$ parameter) associated with the overlap of random states for $t \in [-N/2, N/2]$, i.e.,

$$z_k^2(t) > \eta^2(N), \qquad \forall\, t \in \left[-\frac{N}{2}, \frac{N}{2}\right]. \tag{18}$$

Further, the "optimal" orthonormal basis in which this property is most likely to be present [in terms of maximizing $z_k^2(1)$] was shown to be given by the discrete Fourier transform (DFT) of the energy eigenstates:

$$|C_k\rangle = \frac{1}{\sqrt{N}} \sum_{n=0}^{N-1} e^{-2\pi ikn/N} |E_n\rangle, \tag{19}$$

where the energies are sorted in ascending order. In this case, $z_k^2(t) = z^2(t)$ for all $k$, given in terms of the energy levels by

$$z^2(t) = \left|\frac{1}{N} \sum_{n=0}^{N-1} e^{i(E_n - 2\pi n/N)t}\right|^2. \tag{20}$$

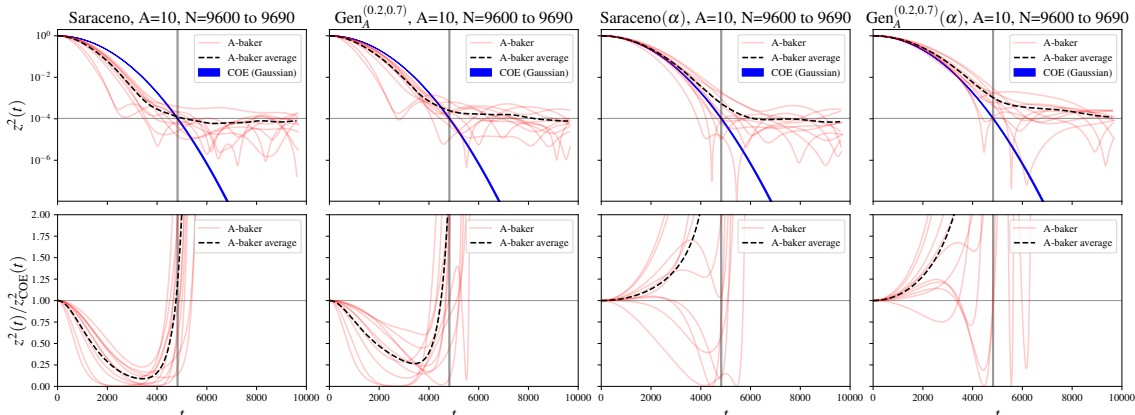

Figure 7: Effect of spectral anomalies on quantum cyclic ergodicity measured in terms of the persistence $z^2(t)$, via four different quantizations of $A = 10$-baker's maps with different kinds of anomalies. The depicted quantizations are Saraceno (no anomalies), $\mathrm{Gen}_A^{(0.2,0.7)}$ (weak reflection and time-reversal anomalies), Saraceno($\alpha$) (strong reflection anomaly) and $\mathrm{Gen}_A^{(0.2,0.7)}(\alpha)$ (strong reflection anomaly, weak time-reversal anomaly). All values of $N$ from 9600 to 9690, in steps of 10, are depicted (translucent red lines) together with the average $z^2(t)$ (top row) and $z^2(t)/z_{\mathrm{COE}}^2(t)$ (bottom row) over these values of $N$ (dashed black line) to observe the statistical trends after averaging out the strong fluctuations with $N$. *Top row*: The persistence $z^2(t)$ given by Eq. (20) is plotted as a function of time $t$ in a log-linear scale, and compared with the COE (Gaussian) curves $z_{\mathrm{COE}}^2(t)$ denoting the ideal behavior of COE statistics given by Eq. (22) over these values of $N$ [up to $O(N^{-1})$ fluctuations, which are not depicted for the COE reference]. The vertical band near the center of each plot depicts the range of $t = N/2$ over the different values of $N$, and the horizontal band depicts $z^2(t) = N^{-1}$ (representing the order of magnitude of $\eta^2(N) = cN^{-1}$), the value reached by the COE (Gaussian) curve at $t = N/2$ (the cutoff time for cyclic ergodicity). *Bottom row*: The ratio $z^2(t)/z_{\mathrm{COE}}^2(t)$ is plotted against $t$ in a linear-linear scale, with the vertical band near the center again depicting the range of $t = N/2$, while the horizontal line near the center depicts a unit ratio, i.e., $z^2(t) = z_{\mathrm{COE}}^2(t)$. The rapid increase near $t = N/2$ in these plots represents the onset of $O(N^{-1})$ fluctuations as the dominant behavior of $z^2(t)$ around and beyond this time. These plots appear consistent with quantum cyclic ergodicity of the kind associated with COE [$z^2(t) \geq z_{\mathrm{COE}}^2(t)$ up to $O(N^{-1})$ fluctuations], resulting from strong anomalies (symmetry breaking in long-range measures) but not weak anomalies as explained in the text.

This is the basis we will study numerically.

For "ideal" RMT-like behavior, $z_{\mathrm{RMT}}^2(t) = \exp[-\Delta^2 t^2]$ to leading order [27] (originating in Gaussian spectral fluctuations [67–69]), where (specializing to even $N$ for simplicity)

$$\Delta^2 = 2 \sum_{t=1}^{N/2} \frac{\mathrm{SFF}(t)}{Nt^2}, \tag{21}$$

gives the leading contribution to spectral fluctuations in various measures of long-range spectral rigidity such as the Dyson-Mehta $\Delta_3$ parameter [3] or the related $\Delta^*$ [70] that measure the regularity of the spectrum. For COE, one obtains

$$z_{\mathrm{COE}}^2(t) = e^{-4t^2 \ln N/N^2}. \tag{22}$$

This is guaranteed to exceed $\eta^2(N) = cN^{-1}$ as per Eq. (18) with the slight restriction $|t| < N(1-\epsilon)/2$ (for any small $\epsilon > 0$), showing that each $|C_k\rangle$ in a system with ideal COE statistics "ergodically" visits almost all basis vectors $|C_{k-N/2}\rangle$ through $|C_{k+N/2}\rangle$ in succession. Due to the presence of time-reversal symmetry without separate sectors in COE [1], one cannot here demand ergodicity in the full interval $|t| \leq N/2$ in Eq. (18), corresponding to fully visiting every single basis vector, that is present [27] in CUE or a non-degenerate half of CSE.

In Fig. 7, the persistence $z^2(t)$ in the DFT basis for 4 different quantizations of the $A = 10$-baker map [Saraceno and $\text{Gen}_A^{(0.2,0.7)}$, with and without phases] are compared to the ideal COE persistence, to examine their quantum dynamical ergodicity relative to the behavior of COE [i.e., if $z^2(t) \geq z^2_{\text{COE}}(t) + O(N^{-1})$]. We recall that while the unitary reflection symmetry may be broken weakly or strongly in these quantizations, the antiunitary time reversal symmetry is always broken only weakly in the spectral statistics, making COE the appropriate standard for comparison. The choice of $A = 10$ statistically guarantees that the Berry-like phases $\alpha_j$ are generically random, as required for strong anomalies (for instance, in Eq. (15)); in contrast, $A = 2$ has only one independent phase and may show a significant dependence on this phase as seen in Fig. 10(g). Further, due to atypical fluctuations with varying $N$ in the level statistics of baker maps [see, e.g., Fig. 8 and Fig. 10(a)-(f)], noted as far back as Ref. [49], we consider statistical trends over 10 adjacent values of $N$, and additionally plot the persistence $z^2(t)$ averaged over these values of $N$ to tame the fluctuations. This is justified for our numerics as $N$ varies only by around 1% in our chosen range. Subsequently, we observe if the average persistence is comparable to (or is greater than) the ideal COE trend to diagnose ergodicity in the presence of a long-range time-reversal symmetry.

The numerical trends are as follows:

1. For Saraceno (no anomalies) and $\text{Gen}_A^{(0.2,0.7)}$ (weak reflection and time-reversal anomalies), $z^2(t)$ remains less than $z^2_{\text{COE}}(t)$ up to random fluctuations consistent with $O(N^{-1})$, showing compatibility with ergodicity-breaking in the presence of a long-range reflection symmetry.

2. For Saraceno with phases (strong reflection anomaly) and $\text{Gen}_A^{(0.2,0.7)}$ with phases (strong reflection anomaly and weak time-reversal anomaly), $z^2(t)$ fluctuates around $z^2_{\text{COE}}(t)$, showing compatibility with the presence of COE-type ergodicity without a long-range reflection symmetry (but with time-reversal indicated by long-range spectral statistics).

Finally, we note that in both the $\text{Gen}_A^{(0.2,0.7)}$ cases (with or without phases), which possess a weak time-reversal anomaly, $z^2(t)$ oscillates around a slightly larger value than in the Saraceno cases (which have an unbroken time-reversal symmetry), though this slight increase does not statistically appear to be sufficient to induce ergodicity without strong anomalies. In fact, this slightly larger value is likely a finite size numerical effect for these values of $N$, stemming from the logarithmic divergence of $\Delta^2$ with $N$ in Eq. (21) for a linear ramp $\text{SFF}(t) \propto t$ leading to a visible numerical contribution from the late-time regime (corresponding to the crossover in Sec. 3.3). However, one can show that in the $N \to \infty$ limit, as long as the SFF appreciably deviates from the early-time trend (due to weak anomalies) only for $|t| \geq cN$ in $\text{SFF}(t)$, the anomalous contribution to $\Delta^2$ is subleading compared to the early-time contribution; it is indeed for a similar reason that COE possesses logarithmically divergent ($\ln N$) spectral fluctuations [3,9,70] despite the SFF deviating from a linear ramp [1] for $t \sim N$. To see this quantitatively, we consider a simplified model with the interval of summation $t \in \mathcal{I} = [1, N/2]$ split into an early-time regime $\mathcal{I}_{\text{UV}} = [1, cN]$ with $\text{SFF}(t) = \alpha t$, and a late-time regime $\mathcal{I}_{\text{IR}} = (cN, N/2]$ with $\text{SFF}(t) = \beta t$ for some $c \ll 1$; in this case, the leading contribution to the logarithmic divergence $\alpha \ln(cN/1)$ comes entirely from the early time region, while the late-time region contributes a subleading term proportional to $\beta \ln[(N/2)/(cN)] = -\beta \ln(2c)$. Nevertheless,

other effects (such as a deviation from a Gaussian profile of $z^2(t)$) are possible at larger $N$, and it would be interesting to explore or rule out such phenomena at values of $N$ at least an order of magnitude larger than the present study.

In summary, our numerics for $N \approx 10^4$ in quantizations of $A$-baker's maps with different manifestations of spectral anomalies appear to be consistent with a direct link between long-range symmetry breaking (strong anomalies) and cyclic ergodicity, with an at best weaker effect of short-range symmetry-breaking (weak anomalies), verifying the analytical connection obtained in Ref. [27] between long-range spectral statistics and quantum dynamical ergodicity.

# 4 Operator symmetries and level spacing statistics

In the remaining sections, we provide further background and details for the results in the previous section. We start with the relation between the quantizations' operator symmetries and the classical map's symmetries.

## 4.1 Operator symmetries

Classifying quantum symmetries corresponding to the classical symmetries in these models is not entirely straightforward. If one can construct a quantum version of the classical symmetry, such as in the Saraceno quantization [50], then one can say that the quantization preserves the corresponding classical symmetry. However, due to the infinite possibilities of quantum operators that can all correspond to same the classical symmetry operator in the limit $\hbar \to 0$ ($N \to \infty$), verifying that a quantization does not commute with any of those operators is much less clear. For this reason, we will discuss a limited version of the possible operator symmetries, and include more detailed analysis in Appendix A. These restricted definitions will still agree with those historically used to describe the symmetries of the Balazs–Voros and Saraceno quantizations [49, 50].

**Quantization on the torus**  To discuss the relation between the classical symmetries and operator symmetries, we first provide more background on the quantization process on the torus. For further details, see [60, 71]. Quantization on the 2-torus associates to each natural number $N \in \mathbb{N}$ and $\theta \in [0, 1)^2$ an $N$-dimensional Hilbert space $\mathcal{H}_N(\theta)$ of quantum states. The parameter $\theta = (\theta_1, \theta_2)$ sets the quasiperiodicity requirement in position and momentum as follows. Letting $S(q, p) = e^{i(pQ - qP)/\hbar}$ denote the phase space translation operators, then the Hilbert space $\mathcal{H}_N(\theta)$ is associated with states $\psi$ on $\mathbb{R}$ satisfying

$$S(1, 0)\psi = e^{-2\pi i \theta_1}\psi, \qquad S(0, 1)\psi = e^{2\pi i \theta_2}\psi,$$

for $\theta = (\theta_1, \theta_2)$. Recall the Balazs–Voros quantization corresponds to the case $\theta_1 = \theta_2 = 0$ which describes periodic states, while the Saraceno quantization corresponds to $\theta_1 = \theta_2 = 1/2$ which describes antiperiodic states. The generic quantization $\text{Gen}_A^{\theta_1, \theta_2}$ corresponds to the quasiperiodic conditions described by $\theta = (\theta_1, \theta_2)$. The main consideration we need for different $\theta$ is that position representation states $|n\rangle$ and momentum representation states $|k\rangle$ are related via the generalized discrete Fourier transform $\hat{F}_N^{\theta_1, \theta_2}$ as defined in Eq. (5), which depends on $\theta$. This explains why one uses the generalized DFT matrices in the Saraceno and $\text{Gen}_A^{\theta_1, \theta_2}$ quantizations. The generalized DFT matrix relation between position and momentum also implies that operators on $\mathcal{H}_N(\theta)$, which are $N \times N$ matrices, are converted between position and momentum basis via conjugation by $\hat{F}_N^{\theta_1, \theta_2}$ (or its inverse).

The Shor baker quantizations involve several different generalized DFT blocks, but we will associate these quantizations with periodic boundary conditions to match the $\hat{F}_N^{-1}$ factor.

**Reflection symmetry**   Let $B$ be the classical $A$-baker's map, and recall the classical reflection symmetry $R$ in Eq. (2), which maps $(q, p)$ to $(1 - q, 1 - p)$ and satisfies $RBR^{-1} = B$. Its quantum analogue $R_N$ should then reverse, in some way, both the position states $|n\rangle$ and the momentum states $|k\rangle$, and quantizations $\hat{U}_N$ that preserve the reflection symmetry should satisfy $R_N \hat{U}_N R_N^{-1} = \hat{U}_N$.

For the Saraceno quantizations, which we will denote here by $\hat{B}_{N,A}^{\text{Sar}}$, the quantum reflection is $R_N : |x\rangle \mapsto |N - 1 - x\rangle$, which has the same action in momentum space and commutes with $\hat{B}_{N,A}^{\text{Sar}}$ since $R_N = (\hat{F}_N^{\frac{1}{2}, \frac{1}{2}})^2$. One can separate the eigenvalues of $\hat{B}_{N,A}^{\text{Sar}}$ according to whether its corresponding eigenstate is in the $+1$ or $-1$ symmetry sector of $R_N$, and this produces COE level spacing statistics within each symmetry sector, as explained in [50]. (See Fig. 9 for larger $A$.) Additionally, when considering the spectrum as a whole, the two symmetry sectors of the Saraceno quantizations combine to look like that of a direct sum of two COE matrices, indicating that the two symmetry sectors behave essentially as if they are independent of each other.

On the other hand, the Balazs–Voros, generic quasiperiodic, and Shor baker quantizations do not exhibit a clear analogous reflection symmetry. We investigate possible *Fourier* reflection symmetries in Appendix A, and provide numerical plots demonstrating the lack of Fourier reflection symmetry for the non-Saraceno quantizations that we consider (Balazs–Voros, $\text{Gen}_A^{0.2,0.7}$, $\text{Gen}_A^{0,0.5}$, and Shor baker). While this rules out a class of reflection operators coming from the generalized DFT matrices, it does not prohibit the possibility of a different commuting reflection-like operator in the $N \to \infty$ limit. For another approach, in Fig. 16 of Appendix A, we also consider the symmetries of phase space (Husimi) plots of the eigenvectors.

**TR symmetry**   The other classical symmetry is a time reversal (TR) symmetry $T : (q, p) \mapsto (p, q)$, which satisfies $TBT^{-1} = B^{-1}$. Its quantum analogue should act on operators by switching between position and momentum basis, and mapping $i \mapsto -i$, so that quantizations $\hat{U}_N$ (in position basis) preserving TR symmetry should ideally satisfy the antiunitary relation

$$\hat{F}_N^{\theta_1, \theta_2} \hat{U}_N (\hat{F}_N^{\theta_1, \theta_2})^{-1} = (\hat{U}_N^{-1})^*, \tag{23}$$

where $^*$ denotes entrywise complex conjugation. We can define a quantization $\hat{U}_N$ to have an "operator TR symmetry" if it satisfies Eq. (23) for its corresponding boundary conditions $\theta$. However, as for the reflection symmetry, other antiunitary operations with the same classical limit could also be a valid "quantum TR symmetry". For the quantizations we consider, the Balazs–Voros and Saraceno quantizations satisfy Eq. (23), while the generic quasiperiodic quantizations with $\theta_1 \neq \theta_2$ and the Shor baker quantizations do not. The same holds for the phase variant quantizations.

## 4.2 Level spacing statistics

Recall the (normalized) level spacings defined in Eq. (8) are given by $s_i = \frac{N}{2\pi}(\theta_{i+1} - \theta_i)$ for $i \in \mathbb{Z}/N\mathbb{Z}$, where $\theta_i$ are the ordered eigenangles of the $N \times N$ unitary matrix. Recall also the mean gap ratio [44], which is a single statistic computed from the level spacings,

$$\langle \tilde{r} \rangle = \left\langle \min\left(\frac{s_{i+1}}{s_i}, \frac{s_i}{s_{i+1}}\right) \right\rangle_i, \tag{24}$$

where the average is over all $i \in \mathbb{Z}/N\mathbb{Z}$. The mean gap ratios for the standard RMT ensembles in the $N \to \infty$ limit were derived in [61], and for block RMT matrices in [48]. The block RMT matrices are relevant in the presence of discrete symmetries, as one generally needs to separate eigenstates according to the symmetry sector to recover expected non-block RMT

Table 3: Mean gap ratio values for RMT ensembles, from [48, 61].

|  | GOE | 2-block GOE | GUE | 2-block GUE | Poisson |
|---|---|---|---|---|---|
| $\langle \tilde{r} \rangle$ | 0.53590 | 0.423415 | 0.60266 | 0.422085 | 0.38629 |

level statistics. We are primarily concerned with the circular orthogonal ensemble (COE) and circular unitary ensemble (CUE). Since the circular ensembles and Gaussian ensembles have the same local $n$-level correlation functions in the limit $N \to \infty$ [3], we may interchange terms such as "COE level spacings" and "GOE level spacings". We list the values of relevance to our study in Tab. 3.

Here the 2-block GOE matrix means a direct sum of two equal sized, independent GOE matrices, and similarly for the the 2-block GUE matrix.

In general, ones expects that chaotic systems with time reversal (TR) symmetry have GOE/COE spectral statistics, while those without have GUE/CUE statistics. Additionally, one expects the presence of discrete symmetries to produce block-RMT statistics, according to the number of symmetry sectors. As we saw for the $A$-baker's map however, the actual level spacings behavior can be highly variable depending on the particular quantization.

We plot in Fig. 8 the mean gap ratios for the different quantizations over a range of $N \in A\mathbb{N}$. As we saw for specific dimensions $N$ in Figs. 2 and 3, out of all the quantizations in Tab. 2, only the Saraceno quantizations, and the generic quantization $\text{Gen}_{A=2}^{0,0.5}$ (for $A = 2$ only), have mean gap ratio close to that for block COE matrices. We note that there are dips in the mean gap ratio at specific values of $N$, many of which relate to powers of the scaling factor $A$ for the non-phase quantizations [Fig. 8(a),(c)]. For such dimensions the level spacings may look non-RMT (sometimes close to Poisson).

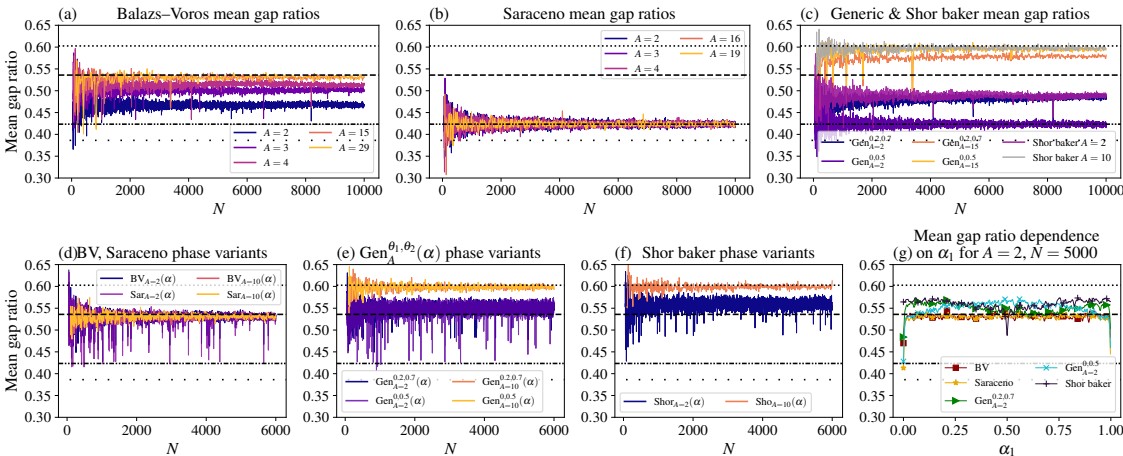

Figure 8: (a)–(f) Mean gap ratios for the different quantizations over $N \in A\mathbb{N}$. The horizontal lines (from top to bottom) are the RMT reference values for GUE (dotted), GOE (dash-dot-dotted), 2-block GOE (dashed), and Poisson (loosely dotted). Some of the phase variant quantizations may share the same random choice of phases. (g) Mean gap ratios for $A = 2$ and $N = 5000$ as a function of the phase $\alpha = (0, \alpha_1)$ for $\alpha_1 \in [0, 1)$ (step size 0.002). Note that $\alpha_1 = 1/2$ corresponds to the standard Shor baker quantization, while $\alpha_1 = 0$ corresponds to the standard versions of the other quantizations.

## 4.3 Approximate symmetry classes for the Balazs–Voros quantization

We now return to the Balazs–Voros-type quantizations of the $A$-baker's map, which we saw have level spacing statistics that can exhibit deviations from RMT and overlook the presence of classical symmetry sectors. We demonstrate how one can obtain roughly COE-like level spacings for the Balazs–Voros quantization in Eq. (3) ($A = 2$) by separating the eigenvalues according to *approximate* symmetry classes of their eigenstates, which was suggested as a possible method in [49]. However, we will see in Sec. 5 using the SFF that this separation still retains significant irregularities.

Recall the reflection operator $R_N : |x\rangle \mapsto |N - x - 1\rangle$ which commutes with the Saraceno quantization and is equal to $(\hat{F}_N^{\frac{1}{2},\frac{1}{2}})^2$. This is the permutation matrix with 1s on the top-right to bottom-left diagonal, which has the trivial block decomposition

$$R_N = \begin{pmatrix} & & & & R_{N/A} \\ & & & R_{N/A} & \\ & & \iddots & & \\ R_{N/A} & & & & \end{pmatrix}.$$

Since it commutes with the Saraceno quantization $\hat{B}_{N,A}^{\mathrm{Sar}}$, this allowed for separating the eigenstates of $\hat{B}_{N,A}^{\mathrm{Sar}}$ according to whether they fall in the $+1$ or $-1$ eigenspace of $R_N$, which recovers RMT spectral statistics.

For the Balazs–Voros quantizations, this suggests considering a similar reflection-like operator, the permutation

$$\tilde{R}_N = (\hat{F}_N^{0,0})^2 = \begin{pmatrix} 1 & 0 & & \cdots & & & 0 \\ 0 & & \cdots & & & 0 & 1 \\ & & & & 0 & 1 & 0 \\ \vdots & & & 0 & 1 & 0 & 0 \\ & \iddots & \iddots & & & & \vdots \\ 0 & 1 & 0 & \cdots & & & 0 \end{pmatrix}, \tag{25}$$

which is a natural reflection candidate (cf. Appendix A) when considering states that are periodic in position and momentum (vs antiperiodic for Saraceno quantizations). The map $\tilde{R}_N$ is equal to $\hat{F}_N^2$ and sends $|x\rangle \mapsto |-x\rangle$ (taken modulo $N$). While $\tilde{R}_N$ does not commute with $\hat{B}_N$, it is in some sense *close* to commuting with $\hat{B}_N$. In particular, we analytically verify in Appendix B that the commutator $[\hat{B}_{N,A}, \tilde{R}_N]$ has only very few non-decaying matrix elements, and numerically plot the Frobenius matrix norm of the commutator in the bottom left corners of Fig. 14(a-c) in Appendix A.

Computing the overlap $\langle \varphi^{(j)} | \tilde{R}_N | \varphi^{(j)} \rangle$ for all eigenvectors $\varphi^{(j)}$ of $\hat{B}_N$, we create the two symmetry classes,

$$\begin{aligned} S_+ &= \{\varphi^{(j)} : \langle \varphi^{(j)} | \tilde{R}_N | \varphi^{(j)} \rangle \geq 0\}, \\ S_- &= \{\varphi^{(j)} : \langle \varphi^{(j)} | \tilde{R}_N | \varphi^{(j)} \rangle < 0\}. \end{aligned} \tag{26}$$

We can then investigate the level spacing statistics within each approximate symmetry class, which are shown (along with those for the exact Saraceno symmetry classes) in Fig. 9.

**Approximate symmetries for $A = 2$** As seen in Fig. 9(c)–(d), for $A = 2$, within a single approximate symmetry class $S_\pm$, the level spacing statistics for the Balazs–Voros quantization look approximately COE. The inner products $\langle \varphi^{(j)} | \tilde{R}_N | \varphi^{(j)} \rangle$ tend to cluster near $-1$ and $1$

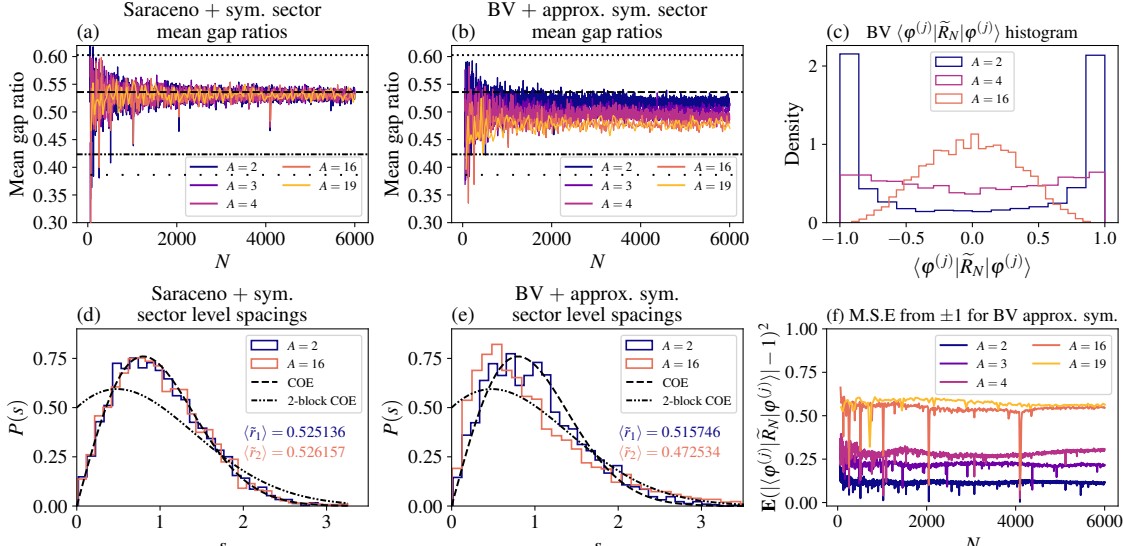

Figure 9: (a), (b) Mean gap ratios for the Saraceno + symmetry sectors and Balazs–Voros + approximate symmetry sectors, for $N \in A\mathbb{N}$ even. (d), (e) Level spacing histograms for the Saraceno + symmetry sector and Balazs–Voros + approximate symmetry sector for $N = 5904$. (c) Balazs–Voros inner product histogram for $N = 5904$ and $A = 2, 4, 16$. The histogram for $A = 2$ shows a strong clustering split between $+1$ and $-1$, but this dichotomy disappears for larger $A$. (f) The mean square error defined in Eq. (27) for the Balazs–Voros quantizations as a function of $N$.

(Fig. 9(e)), suggesting that while not exact, $\tilde{R}_N$ is a fairly good choice of approximate symmetry. Fig. 9(f) plots the quantity,

$$\frac{1}{N} \sum_{j=1}^{N} \left| |\langle \varphi^{(j)} | \tilde{R}_N | \varphi^{(j)} \rangle| - 1 \right|^2 , \tag{27}$$

which is the mean square error of the inner product from $\pm 1$, for eigenstates of $\hat{B}_N$. Other than some outliers that appear somewhat connected to powers of $A$, this error is fairly constant, suggesting that the distribution shape shown for $A = 2$ in Fig. 9(e) is likely representative for other $N$ as well.

We also note that attempting to use the Saraceno reflection operator $R_N : |x\rangle \mapsto |N-x-1\rangle$ here for the Balazs–Voros quantization does not appear to produce any meaningful separation, and the inner products $\langle \varphi^{(j)} | R_N | \varphi^{(j)} \rangle$ are spread within $[-1, 1]$ instead of clustering near $\pm 1$.

**Failure for larger $A$-baker's maps**  For $A \geq 3$, the Saraceno quantizations of the $A$-baker's map continue to commute with the reflection operator $R_N$, and continue to exhibit level spacing statistics that look like a direct sum of two COE matrices. Thus one can try to use an approximate symmetry for the non-symmetrized Balazs–Voros quantizations with $A \geq 3$ as well. Unlike the $A = 2$ case however, the natural approximate symmetry candidate $\tilde{R}_N$ does not produce even an approximately useful separation of eigenstates, as seen in Fig. 9(e). The values $\langle \varphi^{(j)} | \tilde{R}_N | \varphi^{(j)} \rangle$ no longer cluster strongly near $\pm 1$, and separating by the sign of $\langle \varphi^{(j)} | \tilde{R}_N | \varphi^{(j)} \rangle$ does not reproduce RMT-like level statistics (Fig. 9(c)–(d)). Given that the unseparated eigenvalue statistics begin to look more and more like a single COE matrix as $A$ increases, this is not that surprising.

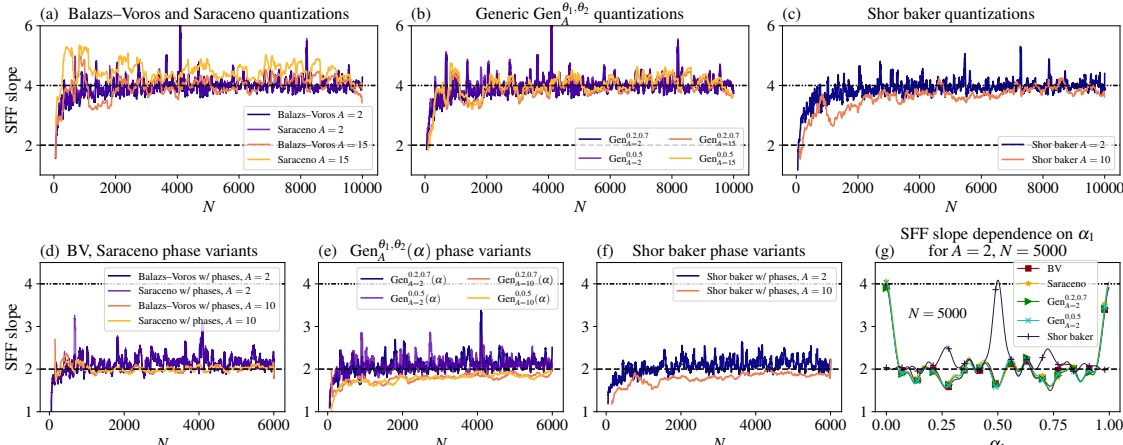

Figure 10: (a)–(c) Averaged early time SFF slope for the standard/phaseless quantizations, plotted as function of $N \in A\mathbb{N}$. The SFF slope values cluster near 4. Outliers (fewer than 1% of points for $A = 2$, and $\sim$5-8% for $A \in \{10, 15\}$) where the least squares slope fitting produced large residuals were removed before averaging. For further details, see Appendix C. (d)–(f) Averaged early time SFF slope for the phase variant quantizations, fewer than 1% of points removed as outliers. The SFF slope values cluster near 2. (g) SFF slope for $A = 2$ and $N = 5000$ as a function of the phase $\alpha \in [0, 1)$ (step size 0.002) for the different types of quantizations. Compare with Fig. 8(g).

## 5 Spectral form factor analysis

In this section, we provide more detailed analysis and plots of the spectral form factor (SFF) and its early time slope. Recall the SFF for an $N \times N$ unitary matrix is given by the formula

$$\text{SFF}(t) = \frac{1}{N}|\text{Tr}(U_N^t)|^2 = \frac{1}{N}\sum_{j,k=1}^{N} e^{it(\theta_j - \theta_k)}, \tag{28}$$

and that we set $\tau = t/N$. The formula for the ensemble-averaged COE form factor [1] is

$$\langle \text{SFF}_{\text{COE}}(\tau) \rangle \equiv \lim_{N \to \infty} \frac{1}{N}\mathbb{E}|\text{Tr}(U_N^t)|^2 = \begin{cases} 2\tau - \tau\log(1 + 2\tau), & \tau \leq 1, \\ 2 - \tau\log\left(\frac{2\tau+1}{2\tau-1}\right), & \tau > 1. \end{cases} \tag{29}$$

For the quantized baker's maps, with no ensemble to average over, we average Eq. (28) at time $t$ with its nearest $2\ell$ neighbors (or from time 1 to $2t - 1$ if $t < \ell$), as described in more detail in Appendix C.

We show plots of the early time SFF slope as function of the dimension $N$ in Fig. 10(a)–(f), corresponding to noisier, more detailed versions of the earlier Fig. 5. In general, the SFF slope computations are noisy, and even the plots in Fig. 10 are averaged over the nearest $\sim$20 neighbors, after removing outliers which did not have a low error slope fit. These outliers amount to only relatively few values of $N$ for each quantization ($< 1\%$ for $A = 2$ quantizations, and $\sim$ 5-8% for $A = 10$ or 15 in Fig. 10). As in Fig. 5, we see in Fig. 10 a clear dichotomy in the SFF slope between the standard phaseless quantizations and the phase variant quantizations. In Fig. 10(g), we also plot the SFF slope for $A = 2$ as a function of the phase parameter $\alpha = (0, \alpha_1)$, similarly as we did for the mean gap ratio in Fig. 8(g).

Next, in Fig. 11(a) we briefly examine the SFF within an individual approximate symmetry class (Sec. 4.3) for the Balazs–Voros 2-baker quantization. We see that while the SFFs appear

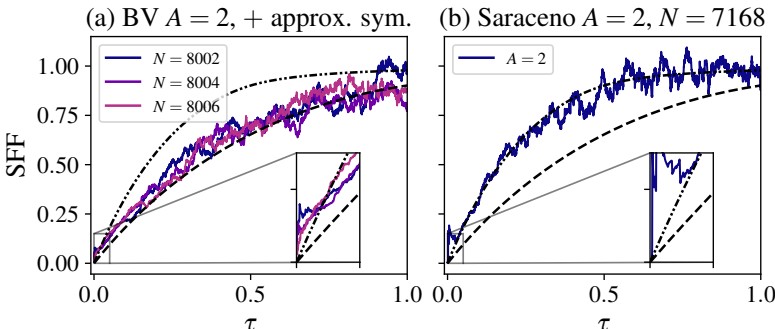

Figure 11: (a) Balazs–Voros SFF for + approximate symmetry classes for $N = 8002, 8004$, and 8006. The behavior for small $\tau$ shows irregularities, even though for larger times it follows the COE SFF. In contrast, the Saraceno $\pm 1$ symmetry classes (not plotted) show single COE-like SFF. (b) Example of bad early time behavior in the SFF, for one of the rare outliers removed before averaging to produce the plots in Fig. 10.

to look COE for moderately sized $\tau$, there are irregularities near $\tau = 0$. Thus while separating by the approximate symmetry class can partially restore level spacing statistics as in Fig. 9, it produces long-range spectral irregularities. In contrast, for the Saraceno quantizations (not shown), the SFF for an individual symmetry class appears to follow the single COE SFF for all $\tau$.

In Fig. 11(b) we also demonstrate a complication with determining the early time SFF slope. For some values of $N$, the SFF may show large early time irregularities. Large enough irregularities which do not have a good least squares fit are considered outliers, and we remove such points prior to averaging and plotting in Figs. 5 and 10.

We note that some of the outliers and noise are products of the averaging methods used to compute the SFF slope. While we do not optimize the averaging methods used, we choose parameters so that it becomes clear whether the slope of the early time SFF is close to 2, corresponding to the SFF for a single COE matrix, or close to 4, corresponding to the SFF for a 2-block COE matrix. Due to this choice of parameters, along with the occasional outliers, computing the SFF slope is not as convenient as computing the gap ratio statistic; however, for these models it proves to be more informative.

# 6 Semiclassical trace formula

In this section, we explain how one derives the semiclassical trace formulas used in Section 3.4. To that end, we must first revisit the classical $A$-baker's map dynamics as used in [59, 62, 63].

## 6.1 Classical dynamics revisited

One particularly useful interpretation of the classical $A$-baker's map is via its symbolic dynamics [51]: Points $(q, p) \in \mathbb{T}^2$ can be identified with infinite base $A$ sequences of symbols $\cdots a_{-2}a_{-1}a_0 \bullet a_1 a_2 \cdots$, where $0.a_1 a_2 \cdots$ is the $A$-ary expansion of $q$, $0.a_0 a_{-1} a_{-2} \cdots$ is the $A$-ary expansion of $p$, and $\bullet$ is a separator distinguishing $p$ from $q$. The classical $A$-baker's map is then the 2-sided Bernoulli left shift,

$$\cdots a_{-2}a_{-1}a_0 \bullet a_1 a_2 \cdots \mapsto \cdots a_{-1}a_0 a_1 \bullet a_2 a_3 \cdots .$$

The composition of the $A$-baker's map with itself $t$ times is then given by $t$ such shifts, or equivalently,

$$q \mapsto A^t q - a_1 \cdots a_t \,,$$
$$p \mapsto A^{-t}(p + a_t \cdots a_1) \,,$$

where digit expressions like $a_1 \cdots a_t$ represent the value when viewed as a base $A$ number, $a_1 \cdots a_t = \sum_{j=1}^{t} a_j A^{t-j}$. The length $t$ periodic orbits of the $A$-baker's map are then seen to be given by $A$-ary expansions of the form $\cdots v v \cdot v v \cdots$ for any length $t$ $A$-ary string $v = a_1 \cdots a_t$. This corresponds to points,

$$q = \frac{v}{A^t - 1} \,, \qquad p = \frac{\bar{v}}{A^t - 1} \,,$$

where $\bar{v} = a_t \cdots a_1$ denotes the $A$-ary reversal of $v$.

As determined in [62, 63], the classical action $S_v$ of a point $v$ is

$$S_v = \frac{v \bar{v}}{A^t - 1} \,. \tag{30}$$

Taken modulo 1, one has

$$S_v = S_{\bar{v}} = S_{R(v)} = S_{R(\bar{v})} \,, \tag{31}$$

where $R(v)$ is the base $A$ reflection operator $R(v) = A^t - 1 - v$. The reflection operator $R$ acts on the expansion $v = a_1 \cdots a_t$ by mapping each digit $a_j$ to the digit $A - 1 - a_j$.

## 6.2 Periodic orbit theory for the generic quantizations

For the semiclassical analysis, we will utilize a mixed basis representation of the quantizations as in [59, 62, 63]. The generic quantization Eq. (4) is written in the position basis, acting on position states $|n\rangle$ and returning states expressed in the position basis. To express a position basis quantization $\hat{U}_{N,\text{pos}}$ in the momentum basis, one takes $\hat{U}_{N,\text{mom}} = \hat{F}_N^{\theta_1,\theta_2} \hat{U}_{N,\text{pos}} (\hat{F}_N^{\theta_1,\theta_2})^{-1}$. For the *mixed* basis quantization, one takes $\hat{U}_{N,\text{mix}} = \hat{F}_N^{\theta_1,\theta_2} \hat{U}_{N,\text{pos}}$, which now acts on position states $|n\rangle$ on the right and momentum dual states $\langle k|$ on the left. Due to the structure of all the quantizations we consider, the mixed basis quantization has a simple block DFT structure. In what follows, quantizations with the subscript "mix" will denote the representation in the mixed basis.

The generic quantization Eq. (4) of the $A$-baker's map $B$ has the simple block diagonal mixed basis representation,

$$\hat{U}_{N,\text{mix}}(\alpha) = \bigoplus_{j=0}^{A-1} e^{2\pi i \alpha_j} \hat{F}_{N/A}^{\theta_1,\theta_2} \,.$$

The classical $t$-step $A$-baker's map $B^t$ can be quantized in a similar way. Letting $v_n = \lfloor A^t n/N \rfloor$, which identifies the length $t$ $A$-ary string corresponding to $n/N$, the corresponding quantization for $B^t$ is, in mixed basis,

$$\langle k| \hat{U}_{\text{mix}}^{(t)}(\alpha) |n\rangle = \delta_{v_n \bar{v}_k} \langle k - \overline{v_n} N/A^t | \hat{F}_{N/A^t}^{\theta_1,\theta_2} | n - v_n N/A^t \rangle e^{2\pi i \sum_{j=0}^{A-1} \alpha_j \eta_j(v_n)} \,, \tag{32}$$

where $\eta_j(v)$ denotes the number of $j$'s in the base $A$ expansion of $v$. The $\delta_{v_n \overline{v_k}}$ term specifies where to place the DFT block $\hat{F}_{N/A^t}^{\theta_1,\theta_2}$; it places it in the row $k$ corresponding to the classical $A$-baker's map image of the rectangle $[v_n/A^t] \times [0,1]$, where $[v_n/A^t]$ denotes the interval $[\frac{v_n}{A^t}, \frac{v_n+1}{A^t})$. One can verify Eq. (32) has the correct phase factor involving $\alpha$ by comparing the action on coherent states to that of $\hat{U}_N(\alpha)^t$. A phase $e^{2\pi i \alpha_j}$ is accumulated for every $j$ in $v$, since a current $q$ value of $0.a \cdots$ (written in base $A$) corresponds to choosing the $a$th DFT block.

The $t$-step quantization in Eq. (32) is not identical to the 1-step quantization $\hat{U}_N(\alpha)$ composed $t$ times, but it is an approximation useful for deriving analytical expressions using a periodic orbit expansion [62, 63]. We will refer to the quantization Eq. (32) of the $t$-step map as the $(t)$-step propagator, with parenthesis, to distinguish it from the 1-step quantization composed $t$ times. Using Eq. (32) (with Eq. (D.2)) for the $(t)$-step propagator in the saddle point method described in [63, §4] yields the approximation for $N \to \infty$,

$$\operatorname{tr}\hat{U}^{(t)} \approx \sum_{\nu=0}^{A^t-1} \frac{A^{t/2}}{A^t-1} e^{2\pi i N S_\nu} \exp\left(2\pi i \sum_{j=0}^{A-1} \alpha_j \eta_j(\nu)\right). \tag{33}$$

As we assume $t \to \infty$ (though slowly) in $N$, we can replace $\frac{A^{t/2}}{A^t-1}$ by $\frac{1}{A^{t/2}}$. Each value $\nu$ in the sum in Eq. (33) corresponds to a length $t$ periodic orbit, given by the coordinates $\nu = a_1 \ldots a_t$ in base $A$.

To estimate the SFF $\frac{1}{N}|\operatorname{tr}\hat{U}^{(t)}|^2$, we expand Eq. (33) in a double sum over indices $\nu, \sigma$. Because of the large factor $N$ in the resulting term $e^{2\pi i N(S_\nu - S_\sigma)}$, we ignore any pairs $(\nu, \sigma)$ with $S_\nu \neq S_\sigma$, since they are likely to average out due to the rapid oscillations. This the "diagonal approximation" method in periodic orbit theory [9]. We know that $S_\nu = S_\sigma$ for $\sigma \in \{\nu, \bar{\nu}, R(\nu), R(\bar{\nu})\}$, and also for any $\sigma$ that is a rotation of any of the four above elements. (A periodic orbit $\nu = a_1 \cdots a_t$ is equivalent to the rotated orbit $a_2 \cdots a_t a_1$, and so on.) For most $\nu$, there are thus $4t$ choices of $\sigma$ that we know satisfy $S_\nu = S_\sigma$. We have overcounted for some $\nu$ however, in particular for the $\nu$ that are repetitions of a shorter sequence, or $\nu$ for which $\{\nu, \bar{\nu}, R(\nu), R(\bar{\nu})\}$ contains duplicates. However, we can count that there are only of order $\mathcal{O}(A^{t/2})$ such $\nu$, which is exponentially small compared to the total number $A^t$ for large $t$. Therefore in what follows we can ignore the differences for such $\nu$ since they contribute non-leading order terms.

Assuming the above-described $4t$ values for $\sigma$ are usually or on average the only main orbits with $S_\sigma = S_\nu$, the diagonal approximation (with the symmetries) then yields

$$\frac{1}{N}|\operatorname{tr}\hat{U}^{(t)}|^2 \approx \sum_{\nu=0}^{A^t-1} \frac{t}{NA^t}\left(2 + 2e^{2\pi i \sum_{j=0}^{A-1} \alpha_j[\eta_j(\nu) - \eta_j(R(\nu))]}\right)$$

$$= \frac{2t}{N} + \frac{2t}{NA^t}\sum_{\nu=0}^{A^t-1} e^{2\pi i \sum_{j=0}^{A-1} \eta_j(\nu)(\alpha_j - \alpha_{A-1-j})}$$

$$= \frac{2t}{N} + \frac{2t}{NA^t}\left(\sum_{j=0}^{A-1} \exp\left(2\pi i(\alpha_j - \alpha_{A-1-j})\right)\right)^t, \tag{34}$$

where we used the multinomial expansion to obtain the last line, since

$$\sum_{\nu=0}^{A^t-1} \exp\left(2\pi i \sum_{j=0}^{A-1} \eta_j(\nu)(\alpha_j - \alpha_{A-1-j})\right) = \sum_{\substack{n_0+\cdots+n_{A-1}=t \\ n_j \in \mathbb{N}_0}} \binom{t}{n_0, \ldots, n_{A-1}} \prod_{j=0}^{A-1} \left(e^{2\pi i(\alpha_j - \alpha_{A-1-j})}\right)^{n_j}.$$

In order for the second term of Eq. (34) not to decay against the $A^t$ term in the denominator as $t \to \infty$, we must have $\alpha_j - \alpha_{A-1-j} = c \mod 1$ for a constant $c$ and all $j = 0, \ldots, A-1$, which requires $c = 0$ or $1/2 \mod 1$ by considering $j = k$ and $j = A - 1 - k$. In the case $c = 0$, we obtain $\frac{1}{N}|\operatorname{tr}\hat{U}^{(t)}|^2 \approx \frac{4t}{N}$, giving an SFF slope of 4 at zero. In the latter case $c = 1/2$, we obtain $\frac{1}{N}|\operatorname{tr}\hat{U}^{(t)}|^2 \approx \frac{2t}{N}(1 + (-1)^t)$, giving an average SFF slope (averaged over $t$) of 2 at zero. Thus as stated in Sec. 3, we only obtain an SFF slope of 4 if $\alpha_j = \alpha_{A-1-j}$ for all $j$, and obtain a slope of 2 in all other cases.

## 6.3 Periodic orbit theory for the Shor baker quantizations

Recall the arbitrary phase version of the Shor baker matrices was defined in Tab. 1 as

$$\hat{U}_N(\alpha) = \hat{F}_N^{-1}\left(\bigoplus_{j=0}^{A-1} e^{2\pi i \alpha_j} \hat{F}_{N/A}^{0,-j/A}\right).$$

(35)

In order to estimate the SFF using the periodic orbit expansion, we must identify the correct $t$-step quantization $\hat{U}^{(t)}$ corresponding to $\hat{U}_N(\alpha)$. For simplicity, we first take all block phases $\alpha_j = 0$, since they can be added in at the end. We next need to keep track of the phases of the 1-step propagator, which we do by calculating its action on maximally localized Gaussian-like states (coherent states) $\Psi_{(q_0,p_0),\sigma,\mathbb{T}^2}$ as defined on the torus, see e.g. [39,71,72]. For $j \in \{0, 1, \dots, A-1\}$, let $\frac{j}{A} \le q < \frac{j+1}{A}$, and also assume $q$ is far enough away from the boundaries $\frac{1}{A}\mathbb{Z}$ to avoid diffraction effects near the classical map's discontinuities. Following the calculations in [39, Suppl. Mat. §III], then for

$$\tilde{U}_N := \bigoplus_{j=0}^{A-1} \hat{F}_{N/A}^{0,\beta_j},$$

and $\Psi_{(q_0,p_0),\sigma,\mathbb{T}^2}$ the torus coherent state at $(q_0, p_0)$, we have the evolution

$$\tilde{U}_N \Psi_{(q_0,p_0),\sigma,\mathbb{T}^2} = e^{i\pi N j q_0} e^{i\pi N j(p_0+j)/A} e^{-2\pi i \beta_j p_0} \Psi_{(Aq_0-j, \frac{p_0+j}{A}),\frac{\sigma}{A^2},\mathbb{T}^2} + o(1),$$

(36)

with the error term $o(1)$ as $N \to \infty$, which includes error from an $\mathcal{O}(N^{-1})$ shift in the coherent state center. The phase $e^{-2\pi i \beta_j p_0}$ is the extra phase due to the $\beta_j$. Starting with a $q_0$ corresponding to $\nu = a_1 \cdots a_t$, then after $t$ applications of $\hat{S}_N$, we accumulate the phase (due to the $\beta_j$)

$$\exp\left(-2\pi i \sum_{j=1}^{t-1} \beta_{a_j}\left[\sum_{i=1}^{j-1} \frac{a_i}{A^{j-i}} + \frac{p_0}{A^{j-1}}\right]\right).$$

(37)

The expression in hard brackets $[\cdots]$ is the momentum coordinate just before applying the $j$th iteration. If we write $p_0 = 0.b_1 b_2 \dots$ in base $A$, then at this step the classical infinite binary sequence is $\cdots b_2 b_1 a_1 \cdots a_{j-1} \bullet a_j \cdots a_t \cdots$, which corresponds to the aforementioned phase. Taking $\beta_j = -j/A$, then Eq. (37) becomes

$$\exp\left(2\pi i \sum_{j=1}^{t-1} a_j \sum_{i=1}^{j-1} \frac{a_i}{A^{j-i+1}}\right) e^{2\pi i \nu p_0/A^t}.$$

(38)

Next we assume the $t$-step propagator $\hat{U}^{(t)}$ is of the form $\hat{F}_N^{-1}\hat{U}_{\text{mix}}^{(t)}$ with $\langle k|\hat{U}_{\text{mix}}^{(t)}|n\rangle = \delta_{\nu_n \bar{\nu}_k} \hat{F}_{N/A^t}^{0,b(\nu)} e^{-2\pi i \psi(\nu)}$ for some $b(\nu)$ and $\psi(\nu)$. As in Eq. (36), the $b(\nu)$ term will produce an extra phase $e^{-2\pi i b(\nu)p_0}$. Comparing this to Eq. (38) leads to the relations $b(\nu) = -\nu/A^t$ and $\psi(\nu) = \phi(\nu)/A$. Adding in the $\alpha_j$ phases then yields the $(t)$-step propagator for Eq. (35) in mixed basis as

$$\langle k|\hat{U}_{\text{mix}}^{(t)}(\alpha)|n\rangle = \delta_{\nu_n \bar{\nu}_k} \hat{F}_{N/A^t}^{0,-\nu/A^t} e^{-2\pi i \phi(\nu)/A} e^{2\pi i \sum_{j=0}^{A-1} \alpha_j \eta_j(\nu_n)},$$

(39)

where

$$\phi(\nu) = -\sum_{j=2}^{t} a_j \sum_{i=1}^{j-1} a_i A^{-j+i}.$$

(40)

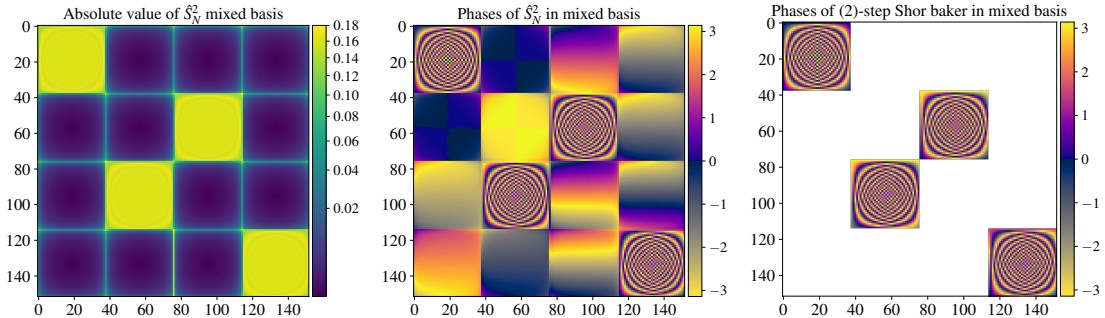

Figure 12: Visual example for Eq. (39). (Left) Plot of the matrix entry sizes of $\hat{S}^2_{N,\text{mix}}$ for $N = 152$. The non-DFT-like blocks have much smaller matrix elements, other than possibly at the block boundary lines. (Center) Phase plot of the entries of $\hat{S}^2_{N,\text{mix}}$. Note the different color patterns in each generalized DFT block (most evident by looking at the four corner areas of each block). This corresponds to different generalized DFT phases and different block phases. (Right) Phase plot of the (2)-step propagator in Eq. (39) with $\alpha = 0$. By carefully considering the different color patterns, one can see they match those of the center plot for $\hat{S}^2_{N,\text{mix}}$.

For visualization purposes, we include graphics below in the style of [59] (which plotted $t$-step propagators for the Saraceno quantization) to visually demonstrate Eq. (39) for the Shor baker quantization with $A = 2$. This involves comparing $\hat{U}^{(t)}_{\text{mix}}$ to the mixed basis propagator $\hat{S}^t_{N,\text{mix}} := \hat{F}_N \hat{S}^t_N$, where

$$\hat{S}_N = \hat{F}_N^{-1} \begin{pmatrix} \hat{F}_{N/2} & \\ & -\hat{F}^{0,-1/2}_{N/2} \end{pmatrix}$$

is the usual $A = 2$ Shor baker quantization. In Figs. 12 and 13, for $t = 2$ and 3, we plot the mixed basis matrix entry sizes and phases of $\hat{S}^t_N$, and observe close agreement with those of the ($t$)-step propagator $\hat{U}^{(t)}_{\text{mix}}$ from Eq. (39) for $A = 2$.

With the stationary phase approximation (see Appendix D for details), Eq. (39) leads to

$$\text{tr}\,\hat{U}^{(t)} \approx \sum_{\nu=0}^{A^t-1} \frac{1}{A^{t/2}} e^{2\pi i N S_\nu} e^{\frac{2\pi i \nu \bar{\nu}}{A^t(A^t-1)}} e^{-2\pi i \frac{\phi(\nu)}{A}} e^{2\pi i \sum_{j=0}^{A-1} \alpha_j \eta_j(\nu)}. \tag{41}$$

From Eq. (40), one can check that $\phi(\nu) = \phi(\bar{\nu})$, and that

$$\phi(R(\nu)) = \phi(\nu) - (A-1)t + A - \frac{1}{A^{t-1}} + 2\sum_{i=1}^{t} a_i - \frac{\nu + \bar{\nu}}{A^{t-1}}. $$

Then we obtain

$$\frac{\nu \bar{\nu}}{A^t(A^t-1)} - \frac{\phi(\nu)}{A} - \left( \frac{R(\nu)R(\bar{\nu})}{A^t(A^t-1)} - \frac{\phi(R(\nu))}{A} \right) = -\left(1 - \frac{1}{A}\right)t + \frac{2}{A}\sum_{i=1}^{t} a_i. \tag{42}$$

Additionally, if $\nu' = a_2 \cdots a_t a_1$ is the 1-step cyclic rotation of $\nu = a_1 \cdots a_t$, then calculation shows that

$$\frac{\phi(\nu')}{A} = \frac{\phi(\nu)}{A} + \frac{a_1}{A^t}(\nu - \bar{\nu}'),$$

so that also using $\frac{\nu' \bar{\nu}'}{A^t-1} = \frac{\nu \bar{\nu}}{A^t-1} + a_1(\nu - \bar{\nu}')$, we obtain

$$\frac{\nu \bar{\nu}}{A^t(A^t-1)} - \frac{\phi(\nu)}{A} = \frac{\nu' \bar{\nu}'}{A^t(A^t-1)} - \frac{\phi(\nu')}{A}. \tag{43}$$

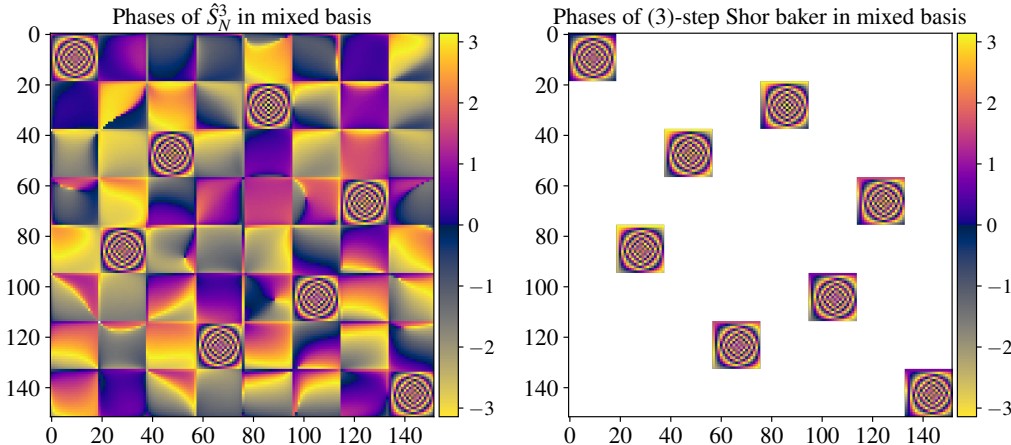

Figure 13: Phase plot equivalents of Fig. 12 for $t = 3$. As in the $t = 2$ case, note the careful agreement between the phases of $\hat{S}^3_{N,\mathrm{mix}}$ and those of the (3)-step propagator from Eq. (39).

Then taking the diagonal approximation (with symmetry factors) to only sum over $\sigma \in \{\nu, \bar{\nu}, R(\nu), R(\bar{\nu})\}$ and their cyclic rotations, yields similarly to Eq. (34),

$$
\frac{1}{N}|\operatorname{tr}\hat{U}^{(t)}|^2 \approx \frac{t}{NA^t}\sum_{\nu=0}^{A^t-1}\left(2 + 2\exp\left(2\pi i \sum_{j=0}^{A-1}\alpha_j\big[\eta_j(\nu)-\eta_j(R(\nu))\big]\right)e^{2\pi i\left[-\left(1-\frac{1}{A}\right)t+\frac{2}{A}\sum_{i=1}^{t}a_i\right]}\right)
$$

$$
= \frac{2t}{N} + \frac{t}{NA^t}\sum_{\nu=0}^{A^t-1}2\exp\left(2\pi i\sum_{j=0}^{A-1}\eta_j(\nu)\Big(\alpha_j-\alpha_{A-1-j}+\frac{2j}{A}\Big)\right)e^{-2\pi i\left(1-\frac{1}{A}\right)t}
$$

$$
= \frac{2t}{N} + \frac{2t}{NA^t}\left(\sum_{j=0}^{A-1}\exp\big(2\pi i(\alpha_j-\alpha_{A-1-j}+2j/A)\big)\right)^t e^{2\pi i t/A}. \tag{44}
$$

In order to have non-decaying second term as $t \to \infty$, we need (modulo 1)

$$
\alpha_j - \alpha_{A-1-j} + \frac{2j}{A} = c, \qquad \forall j = 0,\dots,A-1.
$$

By considering $j = k$ and $j = A-1-k$, we must have $c = -\frac{1}{A}$ or $-\frac{1}{A}+\frac{1}{2}$ (mod 1). In the former, Eq. (44) becomes $\frac{4t}{N}$, while in the latter it becomes $\frac{2t}{N}(1+(-1)^t)$ which averages to slope 2. Thus we obtain an averaged SFF slope of 4 iff

$$
\alpha_{A-1-j} = \alpha_j + \frac{2j+1}{A} \mod 1, \qquad j = 0,\dots,A-1, \tag{45}
$$

and slope 2 in all other cases.

## 7 Conclusion

We have studied maximally chaotic quantum maps with discrete symmetries that share the same classical limit. Contrary to conventional expectations for the correspondence between discrete symmetries and spectral statistics [10,45–48], we demonstrated that short-range spectral statistics in these models generically fail to identify discrete symmetries (weak anomalies),

while long-range spectral statistics also violate these expectations in the presence of phases (strong anomalies). However, long-range spectral statistics appear more directly correlated with intrinsic quantum dynamical properties [27] in the Hilbert space. This further reinforces the notion that spectral statistics should ideally be interpreted in terms of intrinsically quantum mechanical properties, while more work is necessary to understand how they connect to macroscopic dynamics, such as in the classical limit, beyond the well-studied case of systems showing close agreement in several measures with RMT [1,3].

One direction to explore, which may be of immediate relevance in the context of many-body statistical mechanics, is whether the introduction of simple phases — as in the case of strong anomalies studied here — could break the commonly observed correspondence [2,22] between "macroscopic" subsystem thermalization behaviors (i.e. in a large subset of particles) and spectral signatures of ergodic phenomena. While our results already formally point to an affirmative answer, given that one can realize quantizations of $A$-baker's maps as many-body Floquet quantum circuits using the quantum Fourier transform and phase gates [52,53] (with the classical $N \to \infty$ limit then corresponding to the thermodynamic limit of, e.g., many qubits), it would nevertheless be illuminating to understand the mechanisms involved (such as Berry-like phases) in a more natural setting of an interacting many-body system that does not necessarily model a classically chaotic map.

# Acknowledgments

We thank Abu Musa Patoary for useful discussions.

**Funding information**   This work was supported by the U.S. Department of Energy, Office of Science, Basic Energy Sciences under Award No. DE-SC0001911 and the Simons Foundation. The authors acknowledge the University of Maryland supercomputing resources (https://hpcc.umd.edu) made available for conducting the research reported in this paper.

# A   Reflection commutators

In this section, we provide numerical evidence that the (generic) generic quasiperiodic and Shor baker quantizations do not have a Fourier reflection symmetry, as defined below. We also provide numerical plots demonstrating symmetries of various eigenvectors.

We will say that a quantization $\hat{U}_N$ has a "Fourier reflection symmetry" if $\hat{U}_N$ commutes with some $\tilde{R}_N^{\omega_1,\omega_2} := (\hat{F}_N^{\omega_1,\omega_2})^2$, for $(\omega_1, \omega_2) \in [0,1)^2$, for each $N \in A\mathbb{N}$. Interestingly enough, there is a generic quantization $\mathrm{Gen}_{A=2}^{0.5,0}$ that does not commute with its "natural" reflection candidate $\tilde{R}_N^{0.5,0}$, but does commute with $\tilde{R}_N^{0,0}$, and so counts as possessing a Fourier reflection symmetry. As discussed in Sec. 4.1, these Fourier reflection symmetries are only a small subset of all possible quantum reflection operators.

Letting $\hat{B}_{A,N}^{\theta_1,\theta_2}$ be the generic quasiperiodic quantization for the $A$-baker's map, we plot the Frobenius matrix norm for a variety of commutators $[\hat{B}_{N,A}^{\theta_1,\theta_2}, \tilde{R}_N^{\omega_1,\omega_2}]$ in Fig. 14. It appears that the Balazs–Voros quantization, most generic quasiperiodic quantizations, and the Shor baker quantization have nonzero commutators and do not possess a Fourier reflection symmetry.

In Fig. 16, we plot the Husimi functions of eigenstates of the various quantizations. The Husimi function is a phase space representation of a vector $v \in \mathbb{C}^N$, defined using the overlap with coherent states. For a precise definition and further background, see [73]. This type of phase space representation was used in [50] to study scarring of the eigenstates of the Sara-

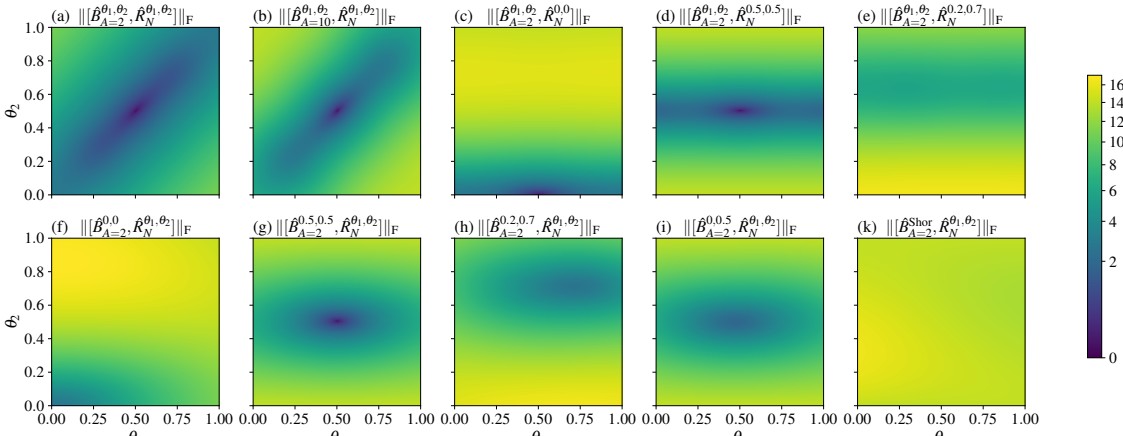

Figure 14: (a)–(e): Plots of the Frobenius matrix norm of the commutator $[\hat{B}_{N,A}^{\theta_1,\theta_2}, \tilde{R}_N^{\omega_1,\omega_2}]$ as a function of $\theta = (\theta_1, \theta_2)$, for: (a)–(b) $\omega = \theta$, (c) $\omega = (0,0)$, (d) $\omega = (0.5, 0.5)$, and (e) $\omega = (0.2, 0.7)$. In all cases $N = 100$. In (a), (b), and (d), the commutator is zero only at $\theta_1 = \theta_2 = 1/2$, which corresponds to the Saraceno quantization. In (c), the $\text{Gen}_{A=2}^{0.5,0}$ quantization is seen (perhaps surprisingly) to commute with $\hat{R}_N^{0,0}$. However, in (e) and for randomly chosen $\omega$, it appears that $\|[\hat{B}_{A,N}^{\theta_1,\theta_2}, \tilde{R}_N^{\omega_1,\omega_2}]\|_F$ is bounded away from zero for all $\theta$. (f)–(k): Plots of the Frobenius matrix norm of the commutator $[\hat{U}_N, \tilde{R}_N^{\theta_1,\theta_2}]$, where $\hat{U}_N$ is a fixed quantization and $\tilde{R}_N^{\theta_1,\theta_2}$ ranges over $\theta \in [0,1)^2$. In all plots except for the Saraceno quantization in (g), the matrix norm appears bounded away from zero, indicating the quantizations should not have a Fourier reflection symmetry. Plots for $A = 10$ appear similar, and plots for the phase variant quantizations also appear bounded away from zero. In all of the above plots, the sampling mesh is size $200 \times 200$.

ceno quantization. Depending on the quantization, the eigenstates may or may not preserve the classical reflection or TR symmetries, which can suggest information about possible quantum symmetries. However, we emphasize that Fig. 16 provides only a rough visual indication of symmetries, of only a select sample of eigenstates, and moreover may contain finite-size effects. Therefore, while the Husimi functions exhibit different symmetries depending on the quantization, they can provide interesting but not conclusive evidence about quantum analogues of the classical symmetries.

# B  Commutator for approximate symmetry

In this section, we analytically check the approximate symmetry $\tilde{R}_N$ introduced in Section 4.3 (Eq. (25)) is in some sense close to commuting with the Balazss–Voros quantization $\hat{B}_{N,A}$. More precisely, for $\hat{B}_{N,A}$ the Balazs–Voros quantization and $\tilde{R}_N = \tilde{R}_N^{0,0}$, we show the only possible large matrix elements $\langle x|[\hat{B}_{N,A}, \tilde{R}_N]|y\rangle$ of the commutator $[\hat{B}_{N,A}, \tilde{R}_N]$ are those $(x, y)$ with $y \in \frac{N}{A}\mathbb{Z}$ and with $x$ close to 0 or $N$ and not in $A\mathbb{Z}$.

Let $a, b \in \{0, \ldots, A-1\}$ be defined so that $a\frac{N}{A} \leq y < (a+1)\frac{N}{A}$ and $b\frac{N}{A} \leq N - y \mod N < (b+1)\frac{N}{A}$. Using that $\tilde{R}_N|y\rangle = |N-y\rangle$ (taken modulo $N$), direct

evaluation shows,

$$\langle x|[\hat{B}_{N,A},\tilde{R}_N]|y\rangle = \frac{\sqrt{A}}{N}\sum_{m=0}^{N/A-1}\left[e^{2\pi iax/A}e^{2\pi ixm/N}e^{2\pi imyA/N} - e^{2\pi ixb/A}e^{-2\pi ixm/N}e^{-2\pi imyA/N}\right]. \quad \text{(B.1)}$$

First, if $x + yA \in N\mathbb{Z}$, which would prevent geometric summation, then since $A|N$ we must also have $x \in A\mathbb{Z}$. Combined with $x + yA \in N\mathbb{Z}$, then Eq. (B.1) is zero in this case. For $x + yA \notin N\mathbb{Z}$, we can evaluate,

$$\langle x|[\hat{B}_{N,A},\tilde{R}_N]|y\rangle = \frac{\sqrt{A}}{N}\left[e^{2\pi iax/A}\frac{e^{2\pi ix/A}-1}{e^{2\pi ix/N}e^{2\pi iyA/N}-1} - e^{-2\pi ibx/A}\frac{e^{-2\pi ix/A}-1}{e^{-2\pi ix/N}e^{-2\pi iyA/N}-1}\right], \quad \text{(B.2)}$$

which we see is zero if $x \in A\mathbb{Z}$. If $y \in \frac{N}{A}\mathbb{Z}$, then one can check that $a + b \in \{0, A\}$, and we use the bound $|e^{2\pi x/N}e^{2\pi iAy/N}-1| \geq \frac{c}{N}d(x, N\mathbb{Z})$ for a numerical constant $c > 0$. This gives the bound

$$\langle x|[\hat{B}_{N,A},\tilde{R}_N]|y\rangle = \mathcal{O}\left(\frac{\sqrt{A}}{d(x, N\mathbb{Z})}\right), \quad \text{(B.3)}$$

which thus allows large commutator matrix elements for the $A$ values of $y \in \frac{N}{A}\mathbb{Z} \cap [0, N-1]$ and $x$ close to 0 or $N$ (and not in $A\mathbb{Z}$).

If $y \notin \frac{N}{A}\mathbb{Z}$, then one can check $a + b = A - 1$, and we obtain from Eq. (B.2) that

$$\langle x|[\hat{B}_{N,A},\tilde{R}_N]|y\rangle = \frac{\sqrt{A}}{N}e^{2\pi iax/A}(1 - e^{2\pi ix/A}) = \mathcal{O}\left(\frac{\sqrt{A}}{N}\right),$$

which is small. Thus the only possible large matrix elements of the commutator $[\hat{B}_{N,A},\tilde{R}_N]$ are those $(x, y)$ from Eq. (B.3) with $y \in \frac{N}{A}\mathbb{Z}$ and $x$ close to 0 or $N$ and not in $A\mathbb{Z}$.

## C  Details for the computation of the early time SFF slope

In this section, we provide the details for our numerical computations of the early time SFF slope. Examples of RMT behavior and (rare) bad early time behavior are shown in Fig. 15.

1. We averaged the SFF at time $t$ with its nearest $2\ell$ neighbors (or up to time $2t - 1$ if $t < \ell$), with $\ell = 20$ for $N < 1000$ and $\ell = 40$ for $N \geq 1000$. The choice of averaging to time $2t - 1$ for $t < \ell$ keeps the averaging symmetric about $t$.

2. We took the first $f$ points of the above averaged SFF, where $f = 20$ for $N < 1000$, $f = 40$ for $1000 \leq N < 5000$, and $f = 60$ for $N \geq 5000$, and ran a least squares fit for a line through the origin to get the best slope. We also retained the scaled residual error, which is the residual error when running the least squares fit for $x \in [\![1 : f]\!]$ and $y = N\,\text{SFF}(x)$.

3. We removed all "outliers" which had scaled residual error over 100 (or 400 for $A = 15$, to make sure not too many points were removed). We then averaged the slopes among points within 10 units away (ignoring outliers) and plotted the resulting slopes. We note that the removed outlier points are not necessarily those with an outlier SFF slope value, but just those for which the least squares fit did not work well.

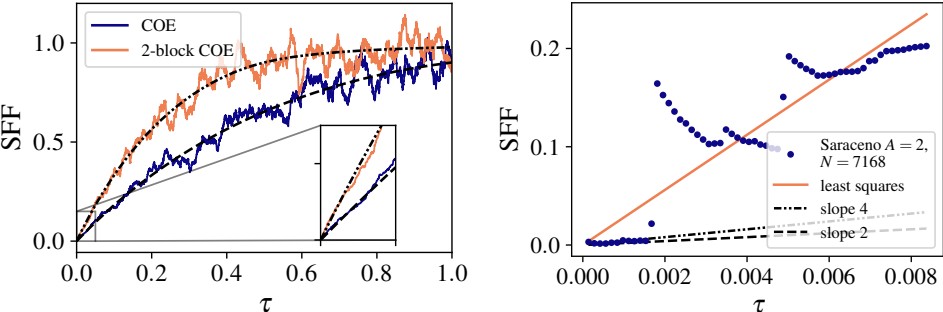

Figure 15: (Left) SFF for random instances of a COE and a 2-block COE matrix for reference, with $N = 9690$ and $\ell = 100$. There is a clear distinction between COE and 2-block COE with this averaging method, which in particular identifies the slope near 0. (Right) Example of the least squares fit for a removed outlier of the Saraceno $A = 2$ quantization, $N = 7168$ (plotted for longer times in Fig. 11(b)). Removed outliers amount to only 0.86% of the values of $N \in 2\mathbb{N}$ considered for this quantization in Fig. 10(b).

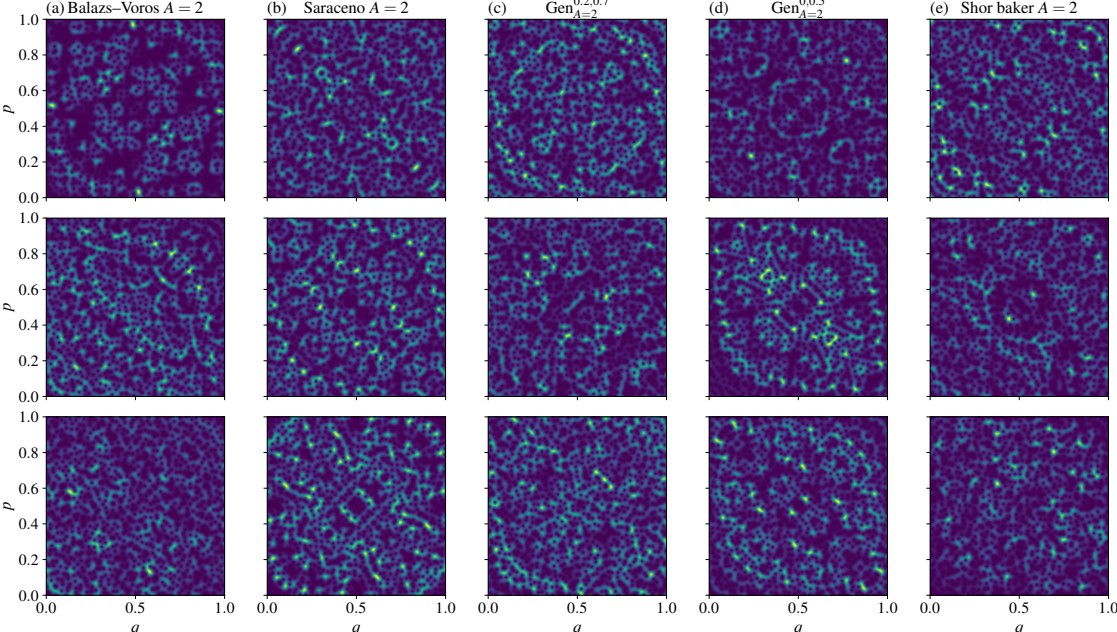

Figure 16: Husimi (phase space) plots for eigenvectors of the various quantizations for $N = 1000$ and mesh size $300 \times 300$, arranged by column. A reflection symmetry across the line $p = q$ corresponds to the classical TR symmetry $(q, p) \mapsto (p, q)$, while a reflection symmetry across the line $p = 1 - q$ corresponds to the classical reflection symmetry $(q, p) \mapsto (1 - q, 1 - p)$. While all quantizations have some eigenvectors that appear to preserve both symmetries (top row), it appears the quantizations that do not have a clear quantum analogue of the classical symmetries can have eigenvectors that break a symmetry (middle and bottom rows of columns (a), (c), and (e)). Of the eigenvectors sampled for the $\text{Gen}_{A=2}^{0,0.5}$ quantization in column (d), however, they appear to generally preserve both classical symmetries.

# D  Shor baker matrix stationary phase approximation

We provide more details for adapting the saddle point method from [63] to the Shor baker quantizations, which we recall involve several different generalized DFT blocks. The resulting extra phase factors in Eq. (D.1) below will be important for the analysis. We start with the $t$-step quantization $\hat{U}^{(t)}_{\text{mix}}$ in Eq. (39), for simplicity with block phases $\alpha_j = 0$ since they can be added in later. The nonzero blocks in $\hat{U}^{(t)}_{\text{mix}}$ correspond to coordinates $(n, k)$ with $\nu_n = \bar{\nu}_k$. Equivalently, picking a $\nu$, then there is the block where $\frac{N\nu}{A^t} \leq n < \frac{N(\nu+1)}{A^t}$ and $\frac{N\bar{\nu}}{A^t} \leq k < \frac{N(\bar{\nu}+1)}{A^t}$. For these coordinates,

$$
\begin{aligned}
\langle k|\hat{U}^{(t)}_{\text{mix}}|n\rangle &= \langle k - \bar{\nu}N/A^t|\hat{F}^{0,-\frac{\nu}{A^t}}_{N/A^t}|n - \nu N/A^t\rangle e^{-2\pi i \phi(\nu)/A} \\
&= \langle k - \bar{\nu}N/A^t|\hat{F}^{0,0}_{N/A^t}|n - \nu N/A^t\rangle e^{2\pi i k\nu/N} e^{-2\pi i \nu\bar{\nu}/A^t} e^{-2\pi i \phi(\nu)/A}.
\end{aligned}
\tag{D.1}
$$

Letting $F_\nu(q, p) = A^t pq - \nu p - \bar{\nu}q$ be the classical generating function as in [63], there is the relation for $q = (n + \theta_2)/N$ and $p = (k + \theta_1)/N$,

$$
\langle k - \bar{\nu}N/A^t|\hat{F}^{\theta_1,\theta_2}_{N/A^t}|n - \nu N/A^t\rangle = \frac{A^{t/2}}{N^{1/2}} e^{-2\pi i N F_\nu(q,p)}.
\tag{D.2}
$$

Since we work with periodic boundary conditions for the Shor baker quantizations, we take $\theta_1 = \theta_2 = 0$. Allowing interpolation to move to continuous coordinates $q$ and $p$, Eq. (D.1) then becomes

$$
\langle p|\hat{U}^{(t)}_{\text{mix}}|q\rangle \approx \frac{A^{t/2}}{N^{1/2}} e^{-2\pi i N F_\nu(q,p)} e^{2\pi i p\nu} e^{-2\pi i \nu\bar{\nu}/A^t} e^{-2\pi i \phi(\nu)/A}.
$$

Preparing for the saddle point approximation as in [63, §4] then yields,

$$
\begin{aligned}
\operatorname{tr}\hat{U}^{(t)} &= \frac{1}{N^{1/2}} \sum_{k,n=0}^{N-1} e^{2\pi i kn/N} \langle k|U^{(t)}_{\text{mix}}|n\rangle \\
&= \frac{1}{N^{1/2}} \sum_{k,n=-\infty}^{\infty} \int_{-\infty}^{\infty} d(Nq) \int_{-\infty}^{\infty} d(Np)\, \chi_{[0,1)}(p)\chi_{[0,1)}(q) e^{2\pi i Npq} \langle p|U^{(t)}_{\text{mix}}|q\rangle \\
&\qquad \times \delta(Nq - n)\delta(Np - k) \\
&\approx N^{3/2} \sum_{\ell,m} \sum_{\nu=0}^{A^t-1} \int_{\frac{\nu}{A^t}}^{\frac{\nu+1}{A^t}} dq \int_{\frac{\bar{\nu}}{A^t}}^{\frac{\bar{\nu}+1}{A^t}} dp\, e^{2\pi i Npq} e^{-2\pi i mNq} e^{-2\pi i \ell Np} \frac{A^{t/2}}{N^{1/2}}\ e^{-2\pi i N F_\nu(q,p)} e^{2\pi i p\nu} \\
&\qquad \times e^{-2\pi i \nu\bar{\nu}/A^t} e^{-2\pi i \phi(\nu)/A} \\
&= N A^{t/2} \sum_{\ell,m} \sum_{\nu=0}^{A^t-1} \int_{\frac{\nu}{A^t}}^{\frac{\nu+1}{A^t}} dq \int_{\frac{\bar{\nu}}{A^t}}^{\frac{\bar{\nu}+1}{A^t}} dp\, \exp\left(2\pi i N[pq - A^t pq + (\nu \quad -\ell)p + (\bar{\nu} - m)q]\right) \\
&\qquad \times e^{2\pi i p\nu} e^{-2\pi i \nu\bar{\nu}/A^t} e^{-2\pi i \phi(\nu)/A}.
\end{aligned}
$$

For $\ell = m = 0$, the stationary point is $q = \frac{\nu}{A^t-1}$, $p = \frac{\bar{\nu}}{A^t-1}$. For other $(\ell, m)$, there are no stationary points in the region of integration, and so ignoring those terms, we thus obtain the stationary phase estimate

$$
\operatorname{tr}\hat{U}^{(t)} \approx \frac{A^{t/2}}{A^t - 1} \sum_{\nu=0}^{A^t-1} e^{2\pi i N S_\nu} e^{\frac{2\pi i \nu\bar{\nu}}{A^t(A^t-1)}} e^{-2\pi i \phi(\nu)/A}.
\tag{D.3}
$$

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
