# Peer review of "Spectral anomalies and broken symmetries in maximally chaotic quantum maps"

_SciPost Physics, doi:SciPost Phys. 18, 151 (2025)_

## Round 1 · Referee Report · Anonymous (Referee 1) · 2024-9-1

Strengths

1. The anomalies in level statistics of quantized A-baker's maps are studied comprehensively, yielding a thorough understanding of the phenomenon.

2. Short-time behavior of the spectral form factor (SFF) is not only observed numerically but also analytically explained with a semiclassical periodic orbit expansion.

3. Deviations from random matrix theory are traced back to approximate symmetries of the quantized maps, enriching the understanding of the phenomenon.

4. Spectral anomalies are linked with the dynamics under the considered quantized maps and understood in terms of quantum cyclic ergodicity.

Weaknesses

1. The manuscript is not easy to follow, it's readability could be improved (see Requested changes below).

2. The notions of the macroscopic symmetry of the classical map and the quantum symmetry of the quantized map are not clearly distinguished and sometimes mixed. The manuscript plainly demonstrates that the relation between the two types of symmetries is not direct and depends on the quantization type. The manuscript's narrative rooted in the premise that both types of symmetries are directly related unnecessarily complicates the reasoning (see Requested changes below).

Report

The manuscript "Spectral anomalies and broken symmetries in maximally chaotic quantum maps" by L. Shou, A. Vikram, and V. Galitski investigates eigenvalue statistics of various quantizations of A-baker's maps. The A-baker's maps are classically chaotic and ergodic and possess two macroscopic discrete symmetries: time-reversal symmetry (TR) and reflection symmetry. Various quantizations of these classical maps exhibit different degrees of symmetry even though all such quantizations reproduce the respective A-baker's map in the semiclassical limit $N \to \infty$.

Properties of spectral statistics and their connection with various types of ergodic behavior are fundamental problems in the study of quantum chaos and dynamics of quantum many-body systems. The present manuscript demonstrates that the discrete macroscopic symmetries are only sometimes reflected in the level statistics of the quantized systems. While this phenomenon was noted in earlier studies referenced in the manuscript, the present manuscript addresses this problem systematically and exhaustively, painting a comprehensive picture of the phenomenon. The manuscript thus provides an important step in a previously identified and long-standing research problem, fulfilling the criteria of SciPost Physics. For this reason, I am inclined to recommend the manuscript for publication, which I will happily do when the authors respond to my remarks below concerning the presentation of the results and the associated reasonings.

Requested changes

1. The abstract states, "By studying various quantizations of maximally chaotic maps that break a discrete classical symmetry upon quantization, we demonstrate that this approach can be misleading and fail to detect macroscopic symmetries."
Sec. 1.2: "We aim to illustrate the unreliability of common spectral statistics in identifying discrete symmetries as may be present in quantized chaotic maps or many-body systems."
Sec. 3.1 "Due to the classical TR and reflection symmetries of the classical A-baker's map, one would expect its quantizations to exhibit spectral statistics similar to a 2-block COE matrix (a direct sum of two independent, equal sized COE matrices)."

The statements above show that the authors seem to imply that level statistics of the quantized maps *should* reflect the macroscopic symmetries of the classical map. This is clearly not the case due to the possibility of quantum mechanical breaking of the symmetry, as written explicitly already in lines 54-55, and then e.g. at the very start of Sec. 4.1. A clear separation of the notions of the macroscopic symmetry of the classical map and the quantum symmetry (associated with a commuting operator) of the quantized map is lacking. Highlighting this non-trivial correspondence would allow the authors to present the results more clearly without undermining the significance of the findigns.

2. The authors write, "These violations are striking in the context of the use of spectral statistics to identify discrete symmetries of the time evolution operator. While such diagnostics are effective in a variety of systems exhibiting block RMT behavior [10,45–48], our results show they cannot always be relied upon, even in simple systems with a well-defined classical limit." Ref. 48 concerns quantum symmetries of quantum systems. The results of the present manuscript highlight the non-triviality of the relation between the level statistics and the macroscopic symmetries of the underlying classical map but never demonstrate the incompatibility of the quantum symmetry of the quantized map and the spectral statistics of the system. Hence, the above quote from the manuscript appears to be misleading.

3. Section 3 does not look like an "Overview of results". While page 7 contains a summary of the results, the subsequent subsections provide more and more detailed accounts of the results, and the whole Section is inconsistent. For instance, Sec. 3.1 does not provide definitions of level spacing distribution and the mean gap ratio, even though these quantities are shown in Fig. 2 (the two quantities are defined only in Sec. 4.2. by Eq 22 and 23). Conversely, Sec. 3.2 provides the definition of SFF, Eq. (8), which is then repeated in Sec. 5 as Eq. 28.
While Sec. 3.1 may serve as an overview of results about short-range spectral statistics, Sec. 3.2, 3.3, and 3.4 extend over nine pages, which is definitely too much for an "Overview of results."
The consequence of this particular section structure is that the results are described in a fragmented and hard-to-follow manner. The idea of Sec. 3 as an overview of the results is not materialized properly in the present manuscript version.

4. Lines 552-553: "there are dips in the mean gap ratio at specific values of $N$ , which typically correspond to powers of the slope". The dips in the mean gap ratio value in Fig. 8 appear to be equidistant, and the horizontal scale is linear in $N$. This seems to be in contradiction with the statement of the dips being related to the powers of $A$. Also, the dips are denoted only with a single color; does this mean they are observed only for a single value of $A$?

5. Line 575: would it be possible to quantify the degree of symmetry breaking by the smallness of the Hilbert-Schmidt norm of this commutator?

6. The parameter $A$ is sometimes called the "slope" of the baker's map. Can this name be introduced somewhere in Sec. 2 (so that the distinction between SFF's slope is more apparent)?

7. The study focuses on eight types of quantizations of the baker's map, which are summarized in Table 1. Could Table 1 also contain information about the quantum symmetries of the given quantization, i.e. TR and reflection symmetry (and mention also the approximate symmetry)? This could simplify the reader's job of linking the observed properties of level statistics with the degree of symmetry of the quantum system.

Recommendation

Ask for minor revision

---

## Round 2 · Referee Report · Anonymous (Referee 2) · 2025-1-12

Strengths

See report of referee 1

Weaknesses

I find figure to be a bit too "busy" so it is hard to grasp the full message. Larger figures and/or less data per figure would, in my opinion, be better.

Report

The manuscript addresses an important question of the role of symmetries in different possible quantizations, using the Baker map example. This comprehensive study points out unexpected features in the quantum spectra going beyond standard expectations in quantally chaotic systems. The first referee provided an outstanding report. As I understand their comments were taken into account in the modified version.

I have one question which the authors might wish to address in the final version/proofs. The mean gap ratios for different quantization schemes vary, moreover being dependent on parameter A. Suppose we have the case of two independent superimposed GOE spectra. The prediction of [48] for a mean gap ratio assumes similar densities of states (dimensions of independent GOE matrices). If one of these matrices is much larger than the other the mean gap ratio will be obviously different ( reaching single GOE value when one of the matrices is much larger than the other). May such considerations help to understand Fig.2?

Requested changes

none

Recommendation

Publish (meets expectations and criteria for this Journal)

---

## Round 2 · Referee Report · Anonymous (Referee 1) · 2025-1-27

Report

The authors have convincingly addressed all my remarks by improving the presentation of the results and making the manuscript more straightforward to follow. I believe that the work "Spectral anomalies and broken symmetries in maximally chaotic quantum maps" provides an important step in a previously identified and long-standing research problem, thus fulfilling the criteria of SciPost Physics. For this reason, I recommend publishing the manuscript in its present form in SciPost Physics.

Recommendation

Publish (easily meets expectations and criteria for this Journal; among top 50%)

---

## Round 2 · Author Response

We thank the Referee for their careful reading of our manuscript and detailed feedback, which has significantly helped improve the clarity of our presentation. In particular, we agree that a careful discussion of "macroscopic" symmetries (to the extent that this is possible) is essential to clearly present our results in context. We address this and all other requested changes in our revised manuscript as indicated below.

---

## Round 2 · List of Changes

Here we provide a detailed changelist and response to the Referee's list of requested changes. 1. We thank the Referee for pointing out that the role of macroscopic symmetries was not clearly stated in the previous version. We have now highlighted the nontrivial nature of the correspondence between symmetries and spectral statistics, in particular by explaining why it is essential to consider only macroscopic symmetries, rather than intrinsic quantum symmetries, before Eq. (3). In short, there is no unambiguous way to define quantum symmetries because any quantum map can be expressed in a diagonal structure in the energy eigenbasis with blocks of size $1$. Further, by transforming to a basis where the eigenvectors are real, one can always identify an antiunitary symmetry of any given quantum system. It is only the restriction to macroscopic symmetries (though imprecise) that identifies which blocks are physically relevant. Thus, it does not seem to us that a clear separation between macroscopic symmetries and quantum symmetries is possible, and to our knowledge, has not been made in the literature. Nevertheless, this is not seen as a "practical" obstacle when studying the spectral statistics of individual systems, which is exceedingly common in the literature.

We also explain why these issue may not be transparent in conventional random matrix theory treatments of this problem in a footnote on the same page. This is because in considering random matrix ensembles rather than individual systems, one is able to restrict to quantum symmetries shared by all members of the ensemble rather than the more numerous set of symmetries of an indiviudal system. Studies of individual systems rather than ensembles, such as ours, do not have this luxury, and must face the ambiguity of how to define symmetries.

  1. We have now clarified this in our manuscript as indicated above. By the same token as above, Ref. [48] considers quantum symmetries of random matrix ensembles rather than individual systems, and circumvents the issue of separating macroscopic symmetries and quantum symmetries. However, to the extent that one desires to apply the results of Ref. [48] to identify the "quantum symmetries" of an individual system rather than an ensemble, it is faced with the same ambiguity. Indeed, we agree with the Referee that this ambiguity highlights the nontrivial nature of the connections between symmetries and spectral statistics. Specifically, we show that (1) quantum symmetries cannot be unambiguously associated with spectral statistics without reference to some macroscopic behavior (which has been frequently recognized in studies of many-body quantum chaos), and (2) the correspondence between macroscopic symmetries and spectral statistics is not straightforward even in systems with a well-defined classical limit.

  2. We thank the Referee for this comment. We have renamed the overall section to "Results" to better reflect its lengthier content, and kept the summary at the beginning of Section 3 as the actual overview of results (now Section 3.1 - Overview of results). The rest of Section 3 contains details of the results, which we have now made self-contained by including definitions where relevant, while the subsequent sections focus on the derivations. Respectfully, we feel that this structure allows a reader interested in our main quantitative results to focus on Sec. 3 (or just 3.1 for a qualitative summary), while a more specialized reader interested in reproducing all the details of our numerics/derivations may consult the later sections for this purpose. To address the Referee's specific concerns, we have added the definition of nearest-neighbor level spacings and mean gap ratio to the beginning of Section 3.2, and rephrased the start of Section 4.2 to reflect the changes.

  3. We thank the Referee for noting the lack of clarity in this description. We amended the statement to clarify it was intended only for the non-phase quantizations, and changed it to "many of which relate to powers of the scaling factor $A$ for the non-phase quantizations [Fig. 8(a),(c)]". We note that while some of the large dips may look linear on the plot in e.g. Fig. 8(a), several of these correspond to different values of $A$. In particular, in Fig. 8 (a), it may be easier to see that there are spikes for all colors; however, the ones for $A=2$ are more numerous due to the smaller value of $A$.

  4. We had numerically calculated the Frobenius norms for a variety of commutators in Fig. 14 in Appendix A (also in the previous version). To ensure that an interested reader would not miss this figure, we have added a clearer explanation in the main text (new lines 608-609) that one can see and compare this commutator's norm in the bottom left corners of Fig. 14(a-c).

  5. We have changed the references to $A$ as the "slope" of the map to calling $A$ the "scaling factor" of the map, and added this description near the beginning of Section 2 (line 120).

  6. We thank the Referee for this suggestion. We have added rows to the summary table in Tab. 2 indicating the preserved classical symmetries for the various quantizations, and added an explanation to the caption. We have chosen to add this information in Tab. 2 instead of Tab. 1 due to its close correspondence to spectral statistics. As we write in the new explanation in the caption (see further discussion in Section 4.1), ruling out symmetries is not entirely straightforward, since the commutation relation would need to be ruled out not just for the standard or "obvious" reflection or TR symmetry operators, but also for other operators that can differ by $O(\hbar)$ but still correspond to the classical symmetry in the semiclassical limit. We clarify that the newly added rows in the table fulfill two purposes: identifying the natural or "obvious" symmetry operators, and identifying whether the classical symmetry is reflected in the short range spectral statistics.

---

## Round 3 · Author Response

We thank both Referees for carefully considering our manuscript and recommending publication in SciPost Physics.

Here, we would like to address the two concerns posed by Referee 1 in the second round.

Regarding their question about mean gap ratios for unequal blocks: indeed the Referee is correct in saying that the mean gap ratios would be different from the predictions of Ref. [48] if the blocks are of unequal size. Given *just* figure 2, one could think of interpolating between the gap ratios for 1 and 2 blocks to "fit" the data. But we emphasize that just because a model can fit the data, it should not be regarded as capturing the underlying physics of the problem.

Moreover, one of the key takeaways from our work is that such a simple fit is not even possible in our case. The spectral form factor in Fig. (4) clearly shows that the early time behavior is often very different from the late time behavior (e.g., consistent with 2 full blocks in the top row at early times, but with differing behaviors at late times), and there is no consistent way to assign a number or relative size of blocks at all times. This is to be contrasted with an average between blocks of unequal size, which would just be a weighted average of the corresponding spectral form factors. Also, in Fig. (6), we show that the behavior of different spectral statistics measures is close to an interpolation between a 1 block and 2 block spectrum (for example), which again is a fit to the data but demonstrates that the fit has to be more complicated than blocks of unequal size. On the whole, we show that we can not interpret the "anomalous" gap ratios as emerging merely from differently sized "symmetry blocks" --- they are instead indicative of genuine dynamical anomalies in the system.

We thank the Referee for their feedback regarding readability of the figures. We have enlarged Figures 1, 2, 6, 11, and 15 to be easier to read. However, most of the other figures (such as Figures 3 and 4) are already at the maximum width for the text. We have considered splitting up several of the figures, but feel that this would run the risk of making it more difficult for the reader to understand the organization of figures, since we already have 16 figures. We have made sure the figures are in high resolution pdf, so that they will be easily viewable electronically.

---

## Editorial Decision

published